# NGS-Marker: Robust Native Watermarking for 3D Gaussian Splatting

**Hao Qin[1], Yukai Sun[1], Luyuan Chen[1], Mengxu Lu[1], Feng Zhang[1,2], Ming Kong[1,2], Zhenhong Du[1,2],\* Qiang Zhu[1]\***

[1]Zhejiang University
[2]Zhejiang Key Laboratory of Geographic Information Science
`{haoqin,3220101205,chenluyuan,lumengxu,zfcarnation,zjukongming,`
`duzhenhong,zhuq}@zju.edu.cn`

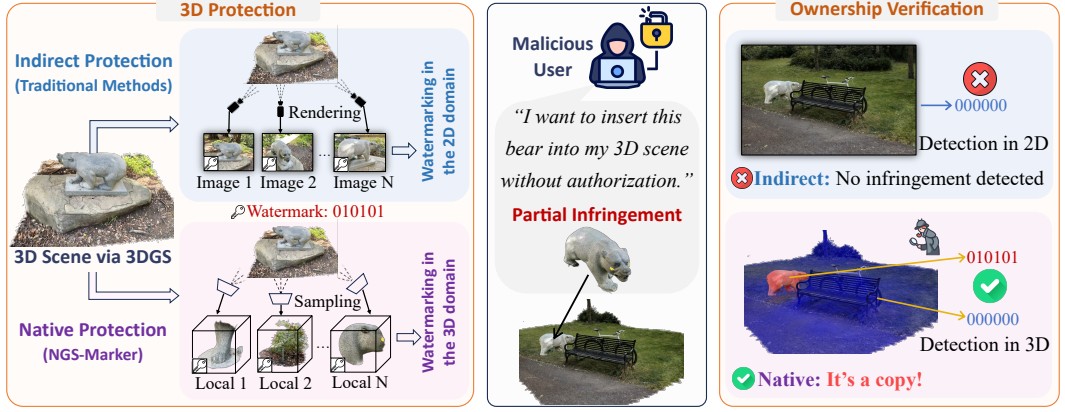

Figure 1: Comparison between native and indirect protection. Due to the explicit nature of 3DGS, malicious users may partially plagiarize protected assets during 3D content creation. When the rendered image distribution shifts significantly, existing indirect protection methods often fail to detect such infringement, whereas NGS-Marker provides reliable protection in these scenarios.

## Abstract

With the rapid development and adoption of 3D Gaussian Splatting (3DGS), the need for effective copyright protection has become increasingly critical. Existing watermarking techniques for 3DGS mainly focus on protecting rendered images via pre-trained decoders, leaving the underlying 3D Gaussian primitives vulnerable to misuse. In particular, they are ineffective against **Partial Infringement**, where an adversary extracts and reuses only a subset of Gaussians. In this paper, we propose **NGS-Marker**, a novel native watermarking framework for 3DGS. It integrates a jointly trained watermark injector and message decoder, and employs a gradient-based progressive injection strategy to ensure full-scene coverage. This enables robust ownership decoding from any local region. We further extend NGS-Marker with hybrid protection (combining native and indirect watermarks) and support for multimodal watermarking. Extensive experiments demonstrate that NGS-Marker effectively defends against partial infringement while offering practical flexibility for real-world deployment.

## 1 Introduction

Recent advances in 3D representations have sparked transformative progress across computer vision and graphics. Among them, 3D Gaussian Splatting (3DGS) Kerbl et al. (2023) has emerged as a mainstream technique due to its efficiency, real-time rendering capabilities, and photo-realistic quality. As 3DGS continues to gain traction in applications such as virtual reality, digital content creation,

---

\*Corresponding author

and robotics Tu et al. (2025); Fu et al. (2025), concerns about copyright protection have become increasingly pressing.

A widely adopted strategy for protecting digital assets involves embedding imperceptible watermarks, which has matured significantly in 2D image domains. Recent efforts have sought to extend such protections to 3D assets by leveraging pre-trained decoders from the image domain Zhu et al. (2025); Luo et al. (2025); Chen et al. (2025). Despite substantial progress, these methods rely on rendered images as intermediate carriers in the watermark embedding and extraction process, thereby protecting Gaussian primitives only indirectly; we refer to this protection paradigm as *indirect watermarking*. The indirect watermarking paradigm suffers from critical limitations: rendering-based extraction inevitably introduces visual degradation, undermining the fidelity of the protected 3D content. More importantly, watermark extraction depends on rendered images, making it vulnerable to appearance shifts such as changes in viewpoint or scene content.

In this paper, we highlight a practical yet underexplored misuse scenario in 3DGS applications: **Partial Infringement**. Due to tile-based rasterization and the independence of Gaussian primitives, 3DGS models are naturally modular, making them susceptible to component-wise extraction and reuse. As illustrated in Figure 1, adversaries can easily isolate objects or characters from a 3D asset and reassemble them into new scenes Chen et al. (2024d). Our study in Section 3 reveals that existing image-based watermarking methods fail under such conditions, where rendering distributions are significantly altered and detection accuracy drops close to random chance.

To avoid the risks associated with extracting watermarks from rendered images, we aim to develop a *native watermarking* framework that dispenses with rendered images as intermediaries and operates directly on Gaussian primitives. A natural approach is to adapt image watermarking techniques Zhu et al. (2018) by training a 3D encoder-decoder network to embed and extract invisible watermarks from Gaussian primitives. However, the number of primitives varies significantly across scenes, which poses a challenge for standard feed-forward networks. Moreover, such a global encoder-decoder strategy remains inadequate against partial infringement.

To this end, we propose NGS-Marker, the first watermarking framework capable of embedding and fully decoding copyright information from local Gaussian primitives. NGS-Marker consists of two key components: a perturbation-based local Gaussian watermark injector and a corresponding extractor. Specifically, we first jointly train both components. The injector takes a fixed-size subset of local Gaussian primitives as input and predicts subtle perturbations to produce watermarked primitives; the extractor then decodes the embedded watermark from these modified inputs. After training, we embed watermarks into the target 3D scene. We find that a naive block-wise application of the injector causes boundary inconsistencies, while repeated random injections lead to cumulative distortion. To address this, we discard the rigid one-shot injection scheme and instead design a soft, progressive optimization strategy. Using the frozen extractor as guidance, we optimize the target scene via gradient descent to ensure uniform watermark distribution and high rendering quality.

NGS-Marker is almost non-conflicting with indirect protection strategies, enabling hybrid protection at both the native and rendering levels to address concerns about image infringement. Additionally, beyond basic bit-based messaging, NGS-Marker supports flexible watermarking with multimodal inputs such as images, allowing identity-aware copyright claims. Experiments on public datasets demonstrate that NGS-Marker not only effectively addresses partial infringement but also exhibits strong robustness against common real-world distortions such as noise, rotation, and sparsification. Our key contributions can be summarized as:

- We investigate the potential risks associated with existing indirect 3DGS protection methods and identify **Partial Infringement**, a prevalent but underexplored misuse scenario in 3D content production.

- We propose **NGS-Marker**, the first framework that enables native protection for local 3DGS. It achieves fine-grained protection of the 3D scene while preserving high rendering quality. Compared with image-based indirect protection approaches, NGS-Marker effectively mitigates partial infringement issues.

- We extend NGS-Marker to enable integration with indirect approaches and support image-based personalized watermarking, demonstrating its potential for applications in privacy-sensitive scenarios. Comprehensive experiments confirm its robustness under various distortions.

## 2 Related Works

**3D Gaussian Splatting**  Recently, 3DGS Kerbl et al. (2023) achieved remarkable success in real-time, high-fidelity rendering, inspiring a series of extensions: Dynamic 3D Gaussians Luiten et al. (2024), 4D Gaussian Splatting Wu et al. (2024a), and Deformable 3D Gaussians Yang et al. (2024) introduce temporal modeling; SuGaR Guédon & Lepetit (2024) and 2D Gaussian Splatting Huang et al. (2024a) improve surface reconstruction; Feature 3DGS Zhou et al. (2024) and LangSplat Qin et al. (2024) enhance scene understanding via feature fields. Meanwhile, 3DGS has been adopted as a foundational representation in tasks such as 3D generation Tang et al. (2024); Chen et al. (2024e); Zou et al. (2024); Tang et al. (2025); Zhang et al. (2025a), editing Fang et al. (2024); Chen et al. (2024c); Wu et al. (2024c), and segmentation Ye et al. (2025); Cen et al. (2025). As its use grows, concerns around intellectual property protection become increasingly prominent.

**Watermarking for 2D Digital Assets**  Embedding imperceptible watermarks into 2D digital assets (e.g., images) for copyright protection has been extensively studied Barni et al. (2001); Cox et al. (2008). Traditional methods use handcrafted signal processing techniques such as DCT, DWT, and SVD to ensure robustness to distortions Cox et al. (1997); Raval & Rege (2003). A major shift occurred with HiDDeN Zhu et al. (2018), which first applied end-to-end neural networks to image watermarking. Subsequent work has improved capacity, generalization, and robustness against adversarial attacks Wan et al. (2022); Tancik et al. (2020), making 2D watermarking increasingly mature and practical Wu et al. (2020); Fernandez et al. (2023); Wen et al. (2023). These developments have also inspired progress in 3D watermarking Jang et al. (2024); Yoo et al. (2022); Luo et al. (2025).

**Watermarking for 3D Digital Assets**  3D assets have diverse structures and require format-specific watermarking. For point clouds, methods perturb coordinates, adjust density, or apply spectral graph techniques to ensure visual fidelity and robustness Li et al. (2021); Yang et al. (2021); Wei et al. (2024). Mesh watermarking leverages vertex shifts, spectral bases, or topology, with deep learning methods enhancing imperceptibility and resistance to attacks like smoothing or simplification Wang et al. (2022); Zhu et al. (2024; 2025). Some studies Narendra et al. (2024); Zaman et al. (2025) have begun to investigate the robustness of watermarks under large-scale cropping; however, they have not thoroughly examined scenarios in which a portion of the protected 3D asset is extracted and embedded into another asset. Radiance fields, encoded implicitly (NeRF) or explicitly (3DGS), are typically watermarked via image modification or radiance regularization Li et al. (2023); Luo et al. (2025; 2023). MarkNeRF Chen et al. (2023) applies image watermarking, while NeRFProtector Song et al. (2024) embeds binary strings via model fine-tuning. Crucially, most current 3DGS watermarking methods Tan et al. (2024); Li et al. (2025) adhere to this pseudo-3D paradigm. These image-space techniques exclusively protect rendered outputs, leaving the underlying native 3D Gaussian primitives susceptible to misuse. GS-Hider Zhang et al. (2024) and WaterGS Guo et al. (2024) achieve the injection of a hidden scene into the global 3DGS, while SecureGS Zhang et al. (2025b) achieves direct protection of Scaffold-GS Lu et al. (2024) by training separate MLPs for each scene. However, native protection mechanisms for local 3DGS data remain largely unexplored.

## 3 Problem Analysis

**Impact of Explicitness on 3D Gaussian Asset Creation**  3DGS explicitly represents the radiance field as a mixture of anisotropic 3D Gaussians $\mathcal{G} = (\mu_i, \alpha_i, s_i, c_i, r_i)_{i=1}^{N}$, where $\mu_i \in R^3$ is the mean, $\alpha_i \in R$ is the opacity, $s_i \in R^3$ is the scale vector, $c_i \in R^c$ is the view-dependent RGB color computed from Spherical Harmonic coefficients, and $r_i \in R^4$ is the rotation quaternion. During rasterization, 3D Gaussians are splatted to screen-space 2D Gaussians following EWA Splatting Zwicker et al. (2002). This process is implemented with a tile-based CUDA rasterizer, which allows real-time differentiable rendering of 3DGS.

Crucially, the independence between Gaussian primitives and the tile-based rasterizer facilitates selective extraction of primitive subsets from arbitrary scenes. With the rapid advancement of 3DGS segmentation techniques Cen et al. (2025), the creation of 3D assets through the free combination of components from multiple sources has become increasingly common Chen et al. (2024d).

**Partial Infringement** We highlight a common yet underexplored misuse scenario in 3DGS termed partial infringement, in which adversaries extract and reuse part of a protected asset without triggering image-based watermark detectors (as shown in Figure 1). To simulate this, we embed a watermark into Scene $\mathcal{A}$ using an indirect method and then transplant a primitive subset into scene $\mathcal{B}$, creating a hybrid scene $\mathcal{B}_{\mathcal{A}}$. Images rendered from $\mathcal{B}_{\mathcal{A}}$ are used to recover the watermark and are compared against the ground-truth message. As shown in Table 1, all tested methods Jang et al. (2025); Huang et al. (2024b); Chen et al. (2025) achieve ~50% accuracy, which is equivalent to random guessing and indicates complete failure. Moreover, reducing embedded bits yields no improvement, implying that the watermark signal introduced via indirect methods is entirely disrupted to the extent that even minimal information cannot be preserved.

Table 1: Results of existing 3DGS watermarking methods under the partial infringement scenario.

| Methods | 8 bits | 16 bits | 24 bits | 32 bits |
|---|---|---|---|---|
| 3D-GSW | 50.35 | 49.17 | 50.52 | 51.60 |
| GaussianMarker | 50.10 | 50.00 | 49.85 | 50.34 |
| GuardSplat | 51.08 | 50.83 | 50.30 | 51.16 |

**Feasibility Analysis** To handle partial infringement and support scenes of arbitrary scale, we propose embedding watermarks into arbitrary local regions of the 3D scene, enabling comprehensive asset protection. This raises a key question: *can a neural network reliably embed and extract watermarks from local sets of Gaussian primitives?* To investigate, we replace the image input in HiDDeN Zhu et al. (2018) with random noise and conduct preliminary experiments in 2D domain. As shown in Figure 2, **the neural network successfully learns to embed and retrieve watermark information from random noise**. Although unstructured, Gaussian primitives exhibit spatial distribution patterns, making them more tractable for neural networks than pure noise. This provides preliminary evidence for our approach's feasibility. In App. N, we provide a more theoretical explanation, grounded in the properties of 3DGS, for why watermarks can be embedded into Gaussian primitives.

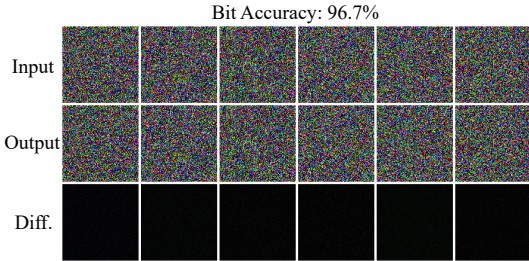

Figure 2: HiDDeN successfully embeds and extracts watermarks from random noise with minimal distortion, demonstrating potential for extension to unstructured data such as 3D Gaussian primitives.

## 4 NGS-MARKER

We propose NGS-Marker, a watermarking strategy designed for 3DGS assets. It embeds watermark information into a source scene $\mathcal{G}^s$ to generate a watermarked version $\mathcal{G}^w$. The method satisfies the following properties: 1) the watermark can be directly verified from $\mathcal{G}^w$ without requiring rendering, 2) the subset of $\mathcal{G}^w$ still reveals the owner's identity, 3) the rendered images of $\mathcal{G}^w$ are visually similar to those of $\mathcal{G}^s$, and 4) the embedded watermark remains robust under common distortions. The overall schematic of NGS-Marker is shown in Figure 3. In the following, we first describe the model training phase and the watermark embedding process in detail. Next, we explain the ownership verification method. Finally, we discuss the image-based watermarking strategy and how NGS-Marker can be coordinated with indirect protection methods.

### 4.1 JOINT TRAINING OF INJECTOR AND EXTRACTOR

We first jointly train a local 3D Gaussian watermark injector and a corresponding message extractor. To embed watermarks into local Gaussians, we apply a perturbation-based strategy by adding small modifications to Gaussian primitives, ensuring the rendering quality remains unaffected. The extractor is then used to recover the embedded message from these perturbed primitives.

As depicted in Figure 3, we first sample a local patch $\tilde{\mathcal{G}}^s$ by randomly selecting $k$ nearest Gaussian primitives from the source scene $\mathcal{G}^s$ based on pairwise center distances. Then, a perturbation feature generator $\mathcal{P}_g$, composed of stacked PointTransformer Wu et al. (2024d) layers, is used to generate the latent perturbation feature $f_d$. The patch $\tilde{\mathcal{G}}^s$ is converted into tokens via a tokenizer (FPS + KNN) and used as the `query` in $\mathcal{P}_g$. The watermark message $\mathcal{M}$ is mapped into a text prompt (with 1 as

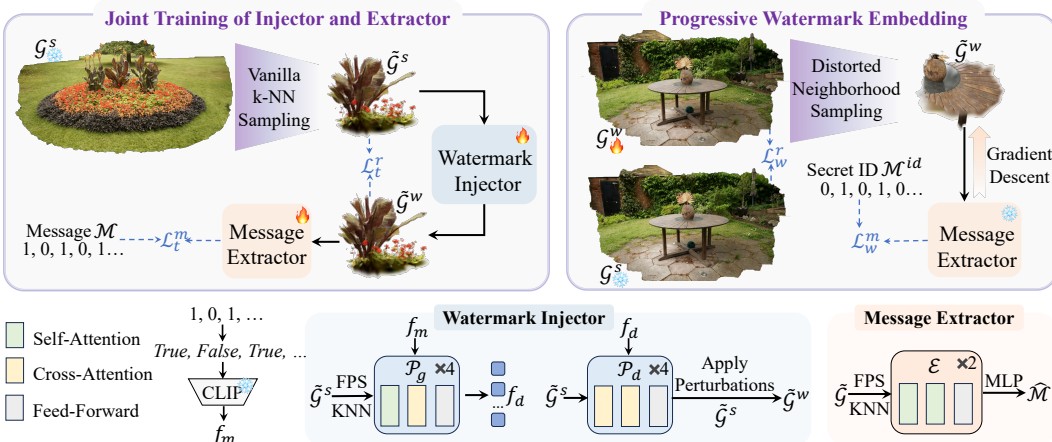

Figure 3: Overview of NGS-Marker. The watermark injector ($\mathcal{P}_g + \mathcal{P}_d$) is jointly trained with the message extractor $\mathcal{E}$. The trained extractor can guide a gradient-based optimization to protect the target 3D scene. Once the private watermark is embedded in $\mathcal{G}^w$, users can verify copyright ownership directly from the native 3D data without rendering.

"*True*" and 0 as "*False*") and encoded using a CLIP text encoder Radford et al. (2021) to obtain $f_m$, which is used as key and value in $\mathcal{P}_g$:

$$f_d = \mathcal{P}_g(\text{tokenizer}(\tilde{\mathcal{G}}^s); \text{CLIP}(\mathcal{M})). \tag{1}$$

To decode the perturbation for each primitive, we design a perturbation decoder $\mathcal{P}_d$, consisting primarily of cross-attention layers. For each Gaussian primitive $\tilde{\mathcal{G}}^s_i$, $\mathcal{P}_d$ takes it as query while using $f_d$ as key and value to predict its corresponding perturbation. Since primitives are processed independently, $\mathcal{P}_d$ can decode all perturbations in parallel. The watermarked patch $\tilde{\mathcal{G}}^w$ is obtained by adding the decoded perturbations to the original primitives:

$$\tilde{\mathcal{G}}^w = \tilde{\mathcal{G}}^s + \mathcal{P}_d(\tilde{\mathcal{G}}^s; f_d). \tag{2}$$

To extract and supervise the watermark embedded in $\tilde{\mathcal{G}}^w$, we design the message extractor $\mathcal{E}$, whose architecture mirrors that of $\mathcal{P}_g$ and consists of stacked PointTransformer layers. $\mathcal{E}$ uses $\tilde{\mathcal{G}}^w$ as the sole input (serving as query, key, and value), and the output of the final PointTransformer layer is passed through an MLP to produce a prediction $\hat{\mathcal{M}}$ with the same dimensionality as the watermark message $\mathcal{M}$. Detailed architectural configurations of $\mathcal{P}_g$, $\mathcal{P}_d$, and $\mathcal{E}$ are provided in App. C.

The injector and extractor are jointly optimized with two objectives: 1) imperceptibility, the rendered appearance of $\tilde{\mathcal{G}}^w$ should resemble that of $\tilde{\mathcal{G}}^s$; and 2) accuracy, the decoded message $\hat{\mathcal{M}}$ should match the original $\mathcal{M}$. The overall loss is defined as:

$$\mathcal{L}_t = \mathcal{L}^r_t + \lambda_t \cdot \mathcal{L}^m_t = \text{MSE}(\mathcal{R}(\tilde{\mathcal{G}}^s, \theta), \mathcal{R}(\tilde{\mathcal{G}}^w, \theta)) + \lambda_t \cdot \text{BCE}(\mathcal{M}, \hat{\mathcal{M}}), \tag{3}$$

where, $\mathcal{R}$ denotes differentiable rendering, $\theta$ is the camera parameters, $\lambda_t$ is a hyperparameter, and MSE and BCE refer to mean squared error and binary cross entropy, respectively.

## 4.2 PROGRESSIVE WATERMARK EMBEDDING

Equipped with the pretrained watermark injector and message extractor, we proceed to protect target scenes. A naive solution is to divide the scene into patches and embed watermarks into each via the local injector. However, this does not ensure uniform distribution across the scene. A more sophisticated method iteratively samples local regions and sequentially embeds the watermark until all areas contain the desired message. Yet, this approach conflicts with the injector's one-shot design: repeated modifications to the same primitives may introduce conflicting perturbations, causing cumulative distortion and degraded rendering quality.

To address this, we adopts a progressive optimization strategy based on the fundamental goal: any randomly sampled local region from the protected scene allows accurate recovery of the identity

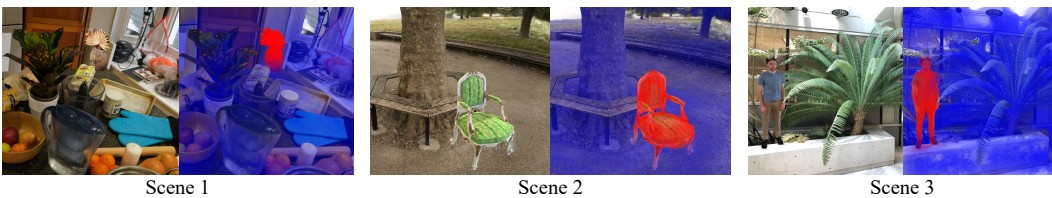

| Scene 1 | Scene 2 | Scene 3 |

Figure 4: Simulated partial infringement scenarios on public datasets. It can be observed that NGS-Marker enables precise and fine-grained copyright protection.

message. As shown in Figure 3, we omit the injector and use the extractor to guide embedding via gradient descent. Specifically, we first randomly sample a patch $\tilde{\mathcal{G}}^w$ consisting of $\delta$ neighboring Gaussian primitives from the target scene $\mathcal{G}^w$. Unlike nearest-neighbor sampling used in training, we introduce some distortions to boost watermark robustness. The sampled patch $\tilde{\mathcal{G}}^w$ is then passed through the extractor $\mathcal{E}$ to produce a predicted message $\hat{\mathcal{M}}$. Finally, we optimize $\mathcal{G}^w$ such that any sampled $\tilde{\mathcal{G}}^w$ enables accurate decoding of the predefined identity message $\mathcal{M}^{id}$.

The objective also consists of two parts: 1) the rendered image of the watermarked scene $\mathcal{G}^w$ should remain visually similar to the source $\mathcal{G}^s$; and 2) the predicted message $\hat{\mathcal{M}}$ should be as close as possible to the target identity $\mathcal{M}^{id}$. The loss function is formulated as:

$$\mathcal{L}_w = \mathcal{L}_w^r + \lambda_w \cdot \mathcal{L}_w^m = \text{MSE}(\mathcal{R}(\mathcal{G}^s, \theta), \mathcal{R}(\mathcal{G}^w, \theta)) + \lambda_w \cdot \text{BCE}(\mathcal{M}^{id}, \hat{\mathcal{M}}). \tag{4}$$

### 4.3 OWNERSHIP VERIFICATION

When suspecting unauthorized use of their assets in a 3D scene, the users can select a suspicious region and input it into the extractor $\mathcal{E}$ to retrieve embedded watermark information, which is then compared with their private ID to determine ownership. To mitigate potential noise in the watermark extracted from a group of primitives, we propose a visualization method for intuitive assessment. We sample multiple primitive groups, determine the most probable owner for each, and assign a unique color associated with the inferred owner onto the SH coefficients of each primitive. This enables intuitive visual localization of regions potentially originating from the user's private assets. Examples demonstrating the applicability of our method are provided in Figure 4 and App. G.

### 4.4 EXTENDED APPLICATIONS

**Cooperating with Indirect Protection Methods** NGS-Marker extracts watermark directly from native 3D Gaussians. However, protecting the rendered images of 3D assets is also important in practice. We can integrate the objectives of both methods to achieve comprehensive protection:

$$\mathcal{L}_{\text{cooperate}} = \mathcal{L}_w + \lambda_{\text{indirect}} \cdot \mathcal{L}_{\text{indirect}}, \tag{5}$$

where $\mathcal{L}_{\text{indirect}}$ denotes the optimization objective of the indirect protection method, which encourages the watermarked renderings to remain decodable while keeping the change in rendering quality small. $\mathcal{L}_w$ is the objective we propose, which enforces that the desired watermark can be decoded from any local Gaussian primitives, again without noticeably degrading rendering quality. In the cooperative protection setting, we optimize these two objectives simultaneously. In Section 5.4, we empirically demonstrate the feasibility of integrating NGS-Marker with indirect protection methods.

**Image-based Watermarking** NGS-Marker is not specifically tailored to binary messages, and in principle, it can handle messages of other modalities. We can replace the CLIP text encoder and the final MLP in the extractor with the encoder-decoder of the target modality, and then fine-tune the model and embed the watermark in target scenes:

$$\mathcal{L}_t' = \mathcal{L}_t^r + \lambda_t' \cdot \mathcal{L}_t^{m'}, \quad \mathcal{L}_w' = \mathcal{L}_w^r + \lambda_w' \cdot \mathcal{L}_w^{m'}, \tag{6}$$

where $\mathcal{L}_t'$ and $\mathcal{L}_w'$ denote losses for model fine-tuning and watermark embedding under the target modality, respectively. $\mathcal{L}_t^{m'}$ and $\mathcal{L}_w^{m'}$ represent the losses for message extraction. In Section 5.5, we empirically show the feasibility of applying NGS-Marker to embed image messages in 3D scenes.

## 5 EXPERIMENTS

### 5.1 EXPERIMENTAL SETUP

**Dataset:** We use standard public 3D datasets to train and evaluate NGS-Marker, with 24 scenes for training and 4 held out for testing. Thanks to the localized watermark injection strategy, a small number of scenes suffices to generate abundant training data. To simulate partial infringement scenarios, we first embed watermark information into test scenes and then extract different subsets of Gaussians from these watermarked scenes. These extracted subsets are inserted into an unwatermarked scene to construct the test dataset. Dataset details are provided in App. D.

**Baselines:** For fair comparison, we evaluate NGS-Marker against five baselines, including three existing 3DGS protection methods and two variants based on our trained local watermark injector: 1) 3D-GSW Jang et al. (2025), 2) GaussianMarker Huang et al. (2024b), 3) GuardSplat Chen et al. (2025), 4) WI-Naive: the target scene is divided into patches, each independently watermarked using the watermark injector, 5) WI-Iterative: a subset of the scene is randomly selected and watermarked, followed by repeated sampling and injection over multiple iterations.

**Implementation Details:** The watermark injector and message extractor are trained on two A100 GPUs with $k = 8192$ and $\lambda_t = 5$ for 150 epochs. Watermark embedding is performed via progressive optimization of the target 3D scene on a single A100 GPU, using $\delta = 8192$ and $\lambda_w = 5$. The number of optimization iterations is adapted to scene complexity and stops once evaluation metrics stabilize. During random sampling, the applied distortions include densification, noise, rotation, dropout, and translation. Unless stated otherwise, the embedded message length is 16 bits.

**Evaluation Metrics:** We evaluate NGS-Marker from three standard aspects of digital watermarking: **(i)** Capacity (Bit-Acc and 3D-Acc): Measures bit-level accuracy and primitive-level accuracy across different message lengths. 3D-Acc is calculated as follows: a Gaussian primitive is randomly selected, and its $\delta$-nearest neighbors are used as input to the message decoder. The decoded message is then compared with the ground truth ID. If the similarity exceeds a fixed threshold $\tau$, the Gaussian is classified as belonging to a protected asset; otherwise, it is not. We set $\tau$ to 75% in all experiments in this paper, and the final accuracy is obtained by averaging the correctness over all test Gaussians. For further discussion on $\tau$, please refer to App. F. **(ii)** Imperceptibility (PSNR, SSIM, LPIPS): Perceptual similarity between rendered images before and after watermarking. **(iii)** Robustness (Gaussian Noise, Rotation, Scaling, Densification, Dropout, Translation): Testing whether the embedded watermark remains decodable under various distortions.

### 5.2 EXPERIMENTAL RESULTS

**Accuracy** We evaluate two metrics: bit accuracy (Bit-Acc), which measures the similarity between the extracted and injected bit sequences, and 3D accuracy (3D-Acc), which reflects the average classification accuracy at the level of individual Gaussians. For Bit-Acc, our method directly decodes information from the Gaussians, whereas other baselines rely on rendered images for watermark extraction. As shown in Table 2, existing public methods achieve near-random performance ($\sim$50%) under partial infringement, demonstrating their inability to cope with this scenario. The two variants using our local watermark injector can recover part of the embedded message, but still fall short compared to NGS-Marker in terms of decoding accuracy.

Since existing methods cannot perform detection at the per-Gaussian level, we report their 3D-Acc as 'N/A'. Notably, GaussianMarker includes a decoder that extracts information directly from Gaussian properties, but it is scene-specific, tied to a fixed watermark, and does not generalize. When Gaussians from a watermarked scene are partially inserted into a new scene, the boundary regions contain a mixture of watermarked and non-watermarked elements, significantly increasing the detection difficulty. As shown in Table 2, NGS-Marker achieves over 95% 3D-Acc, demonstrating strong robustness under mixed watermark conditions. Moreover, we observe that 3D-Acc does not degrade as the number of embedded bits increases. This demonstrates NGS-Marker's ability to maintain high localization accuracy even with longer, more secure watermarks. We hypothesize that although Bit-Acc may drop slightly with longer messages, the overall robustness of the watermark improves, which results in fewer false positives caused by accidental matches. Interestingly, when fewer bits are embedded (*e.g.*, 8 bits), 3D-Acc tends to decrease, likely due to the higher chance that unmarked Gaussians are misclassified as protected assets due to random coincidence.

Table 2: Quantitative comparison with baselines in partial infringement scenarios. Results are reported for 8-, 16-, and 24-bit messages and averaged over all test scenes. 'N/A' indicates that the corresponding method does not support this functionality. Superscripts '*' and '§' denote watermark extraction from rendered images and Gaussian primitives, respectively.

| Methods | 8 bits | | | | 16 bits | | | | 24 bits | | | |
|---|---|---|---|---|---|---|---|---|---|---|---|---|
| | Bit-Acc | 3D-Acc | PSNR/SSIM↑ | LPIPS↓ | Bit-Acc | 3D-Acc | PSNR/SSIM↑ | LPIPS↓ | Bit-Acc | 3D-Acc | PSNR/SSIM↑ | LPIPS↓ |
| 3D-GSW | 50.35* | N/A | 31.88 / 0.978 | 0.027 | 49.17* | N/A | 30.37 / 0.960 | 0.051 | 50.52* | N/A | 29.82 / 0.955 | 0.061 |
| GaussianMarker | 50.10* | N/A | 31.81 / 0.973 | 0.031 | 50.00* | N/A | 30.75 / 0.961 | 0.046 | 49.85* | N/A | 29.69 / 0.950 | 0.057 |
| GuardSplat | 51.08* | N/A | 40.74 / 0.996 | 0.010 | 50.83* | N/A | 39.22 / 0.994 | 0.013 | 50.30* | N/A | 37.96 / 0.991 | 0.022 |
| WI-Naive | 65.32§ | 68.50 | 30.15 / 0.970 | 0.041 | 57.54§ | 55.90 | 27.07 / 0.962 | 0.060 | 54.29§ | 53.70 | 25.36 / 0.911 | 0.093 |
| WI-Iterative | 77.93§ | 80.70 | 26.36 / 0.904 | 0.092 | 69.25§ | 72.30 | 25.54 / 0.886 | 0.105 | 60.38§ | 61.50 | 24.47 / 0.882 | 0.136 |
| **NGS-Marker** | **99.14§** | **95.20** | **41.77 / 0.996** | **0.004** | **97.94§** | **96.60** | **40.17 / 0.995** | **0.007** | **94.68§** | **96.50** | **39.61 / 0.993** | **0.013** |

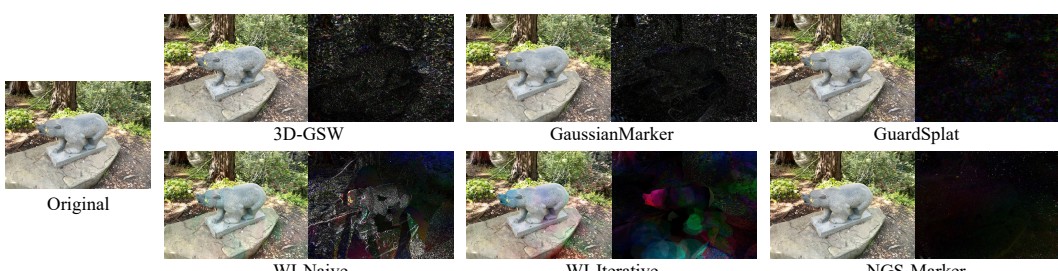

Original    3D-GSW    GaussianMarker    GuardSplat    WI-Naive    WI-Iterative    NGS-Marker

Figure 5: Visual comparisons in the `bear` scene with 16-bit watermark embedding. For better visualization, the difference images are amplified by a factor of 5. It can be seen that our method introduces negligible visual distortion.

**Rendering Quality** Since NGS-Marker does not require extracting watermark information from rendered images, it is theoretically capable of embedding watermarks without compromising rendering quality. In contrast, existing methods inevitably alter the rendered images to encode watermark signals. As shown in Table 2 and Figure 5, we provide both quantitative and qualitative comparisons of rendering quality before and after watermark embedding. While existing approaches are able to preserve visual fidelity to a high degree, our method consistently achieves the best results. Moreover, direct watermark injection using only the local injector significantly degrades rendering quality, further validating the necessity and effectiveness of the progressive optimization strategy.

**Robustness for Distortions** In practical scenarios, watermarked 3D assets are often subject to various distortions. To evaluate the robustness of watermarks embedded by NGS-Marker, we conduct experiments simulating a range of such distortions. We first embed watermarks into the test scenes,

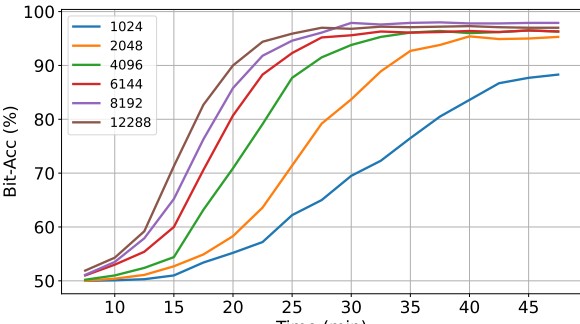

Figure 6: Impact of different $\delta$ values on convergence time and Bit-Acc in `bear` scene. NGS-Marker remains effective even at a high watermarking granularity.

then apply different types of distortions, and finally assess whether the extractor can still recover the watermark from the altered assets. For distortions that degrade rendering quality, such as Gaussian noise, densification, and dropout, we cap distortion intensity to preserve visual integrity of the rendered images, since excessive distortion would typically be avoided in real-world misuse cases. As shown in Table 3, NGS-Marker exhibits strong robustness against most types of plausible distortions. Additionally, because we normalize spatial positions before feeding the local Gaussians into the injector and extractor by scaling all Gaussian centers into a unit sphere, our method is inherently invariant to global scaling and translation of the Gaussians.

Table 3: Quantitative robustness evaluation under various attacks. 'Combined' denotes concurrent exposure to all attack types. Results are averaged over five runs to account for attack randomness.

| Metrics | None | Gaussian Noise ($\sigma$=0.015) | Rotation ($\pm\pi$) | Scaling ($\pm\infty$) | Densification (0%-50%) | Dropout (0%-50%) | Translation ($\pm\infty$) | Combined |
|---|---|---|---|---|---|---|---|---|
| Bit-Acc $\uparrow$ | 98.35 | 97.06 | 98.31 | 98.35 | 97.93 | 97.41 | 98.35 | 96.28 |
| 3D-Acc $\uparrow$ | 97.90 | 96.80 | 97.80 | 97.90 | 97.40 | 97.50 | 97.90 | 96.30 |

Table 5: Results of combining NGS-Marker with indirect protection methods. 'Indirect' refers to indirect watermarking; 'Native' indicates native protection using Equation 4; 'Hybrid' combines both via optimization in Equation 5.

| Type | 3D-GSW | | | | GaussianMarker | | | | GuardSplat | | | |
|---|---|---|---|---|---|---|---|---|---|---|---|---|
| | PSNR/SSIM$\uparrow$ | LPIPS$\downarrow$ | 3D-Acc | Bit-Acc | PSNR/SSIM$\uparrow$ | LPIPS$\downarrow$ | 3D-Acc | Bit-Acc | PSNR/SSIM$\uparrow$ | LPIPS$\downarrow$ | 3D-Acc | Bit-Acc |
| Indirect | 30.37 / 0.960 | 0.051 | N/A | 99.07* | 30.75 / 0.961 | 0.046 | N/A | 98.85* | 39.22 / 0.994 | 0.013 | N/A | 99.51* |
| Native | 40.17 / 0.995 | 0.007 | 97.90 | 98.35§ | 40.17 / 0.995 | 0.007 | 97.90 | 98.35§ | 40.17 / 0.995 | 0.007 | 97.90 | 98.35§ |
| Hybrid | 33.16 / 0.980 | 0.039 | 97.10 | 98.68* 97.59§ | 35.09 / 0.983 | 0.022 | 97.40 | 98.21* 97.84§ | 39.46 / 0.994 | 0.011 | 97.00 | 99.14* 97.38§ |

## 5.3 ABLATION STUDIES

We set the default value of $\delta$ to 8,192; however, in some scenarios, users may wish to detect watermark information from as few Gaussians as possible. To evaluate the effectiveness of our method under varying numbers of local Gaussians, we conduct an ablation study. Using the `bear` scene,

Table 4: Watermarking time for different scenes and the number of primitives contained therein.

| Scene | person | chair | bear | garden |
|---|---|---|---|---|
| Time (min) | 4.0 | 9.3 | 28.7 | 35.2 |
| Number | 42512 | 116713 | 418979 | 588946 |

we test how different $\delta$ values affect the convergence time of watermarking and the final Bit-Acc. As shown in Figure 6, reducing $\delta$ leads to slower convergence and a slight drop in final accuracy. Nevertheless, our method performs reliably when $\delta$ is 2,048 or higher, demonstrating its potential for detecting watermarks in small-scale infringement scenarios.

Additionally, we report the time required to embed watermarks into each test scene, as shown in Table 4. It can be observed that the embedding time is positively correlated with the number of primitives.

## 5.4 COOPERATING WITH INDIRECT PROTECTION METHODS

We employ Equation 5 to examine the feasibility of integrating NGS-Marker with indirect protection methods, with the parameter $\lambda_{\text{indirect}}$ set to 0.1. For 3D-GSW Jang et al. (2025) and Gaussian-Marker Huang et al. (2024b), we first expand or prune the primitives according to their respective original procedures, followed by watermark embedding. As presented in Table 5, NGS-Marker does not exhibit notable conflicts with rendering-based indirect protection methods. Their joint application enables comprehensive protection of 3DGS. More results are shown in App. E.

## 5.5 IMAGE-BASED WATERMARKING

We design a simple experiment to test whether users can inject image-based watermarks into 3D scenes using NGS-Marker. We replace the CLIP text encoder (originally used to encode bit messages) with the CLIP image encoder, and substitute the final MLP in the extractor with an image decoder composed of transposed convolutional layers. All components except the image encoder are fine-tuned. The optimization objectives during fine-tuning and

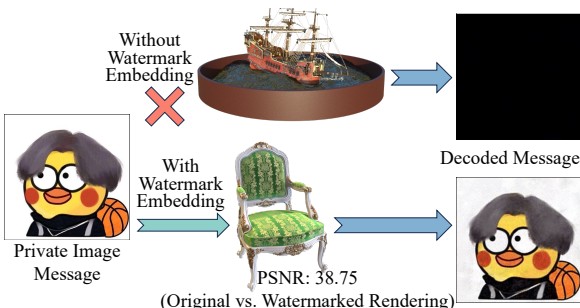

Figure 7: Our method successfully embeds and recovers image watermarks in 3D scenes.

embedding watermarks are:

$$
\begin{aligned}
\mathcal{L}_t^{\text{image}} &= \mathcal{L}_t^r + 5 \cdot \text{MSE}(\hat{\mathcal{M}}^{\text{watermarked}}, \mathcal{M}^I) + 5 \cdot \text{MSE}(\hat{\mathcal{M}}^{\text{original}}, \mathcal{M}^B), \\
\mathcal{L}_w^{\text{image}} &= \mathcal{L}_w^r + 20 \cdot \text{MSE}(\hat{\mathcal{M}}^{\text{watermarked}}, \mathcal{M}^{I_{id}}),
\end{aligned}
\tag{7}
$$

where, $\hat{\mathcal{M}}^{\text{watermarked}}$ and $\hat{\mathcal{M}}^{\text{original}}$ denote the decoded images from watermarked and original Gaussians, respectively. $\mathcal{M}^I$ is the watermark image, and $\mathcal{M}^B$ is a blank image of the same size. We select three images from the internet as target watermarks and fine-tune the model. As shown in Figure 7, we extract and visualize the decoded watermark information from two scenes: a clean scene (`ship`) and a watermarked scene (`chair`). The results show a clear distinction between the two, demonstrating the potential of our method to support image-based watermarking for 3DGS.

## 6 CONCLUSION

We highlight a common but previously underexplored misuse scenario for 3DGS assets, namely **Partial Infringement**. To address this issue, we propose **NGS-Marker**, a novel native watermarking framework tailored for 3DGS. Leveraging carefully designed training and embedding strategies, NGS-Marker achieves efficient and robust protection for 3DGS assets. Furthermore, our experiments demonstrate that NGS-Marker supports multimodal watermark messages and can be integrated with traditional indirect protection techniques, thereby enhancing its practical applicability.

**Limitations and Discussion**   Although NGS-Marker achieves fine-grained and stable protection for 3DGS, it still has some limitations. 1) Similar to other native 3D works Chen et al. (2024a), NGS-Marker relies on 3D encoding/decoding techniques that are less mature compared to 2D. 2) Additionally, since our method theoretically supports arbitrarily large scenes, designing an efficient partitioning and traversal strategy could further improve scalability.

## 7 ACKNOWLEDGEMENTS

This work was supported by the National Natural Science Foundation of China under Grant 42394060 and 42394064.

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

APPENDIX

## A  OVERVIEW OF THE APPENDIX

Here, we supplement some detailed information that is not explained in the main text but may be of interest to the reader, including:

- App. B: Why is "Nativeness" Important;
- App. C: Detailed Model Architecture;
- App. D: Detailed Dataset Information;
- App. E: Cooperating with HiDDeN;
- App. F: Further Discussion on $\tau$;
- App. G: Additional Visualization Examples;
- App. H: Use of LLMs;
- App. I: Time Comparison;
- App. J: Computational Complexity;
- App. K: More Ablation Studies;
- App. L: Robustness Against Adversarial Attacks and 3D Editing;
- App. M: Experimental Results on Complex and Dynamic Scenes;
- App. N: Why Can Gaussian Primitives be Used for Watermark Embedding?

In addition, we provide the Demo.mp4 in the supplementary materials and simultaneously upload source code to the anonymous repository: `https://anonymous.4open.science/r/NGS-Marker/`

**Ethics Statement** This research uses only publicly available datasets that contain no personally identifiable or sensitive information. No human or animal subjects are involved. All experiments are conducted in accordance with institutional and conference ethical guidelines.

**Reproducibility Statement** We provide detailed descriptions of the model architecture and hyperparameter settings in Section 5.1 and App. C. All datasets used are publicly available. Our code will be released to facilitate reproducibility.

## B  WHY IS "NATIVENESS" IMPORTANT

In this work, we propose a copyright protection method tailored for native 3D Gaussian Splatting (3DGS) data. Here, we elaborate on the significance of nativeness in our approach:

**(1) Enhanced robustness**: Watermarks embedded directly into native 3D data exhibit strong resilience against transformations such as viewpoint changes and resolution variations during detection.

**(2) Tight coupling**: The watermark is embedded within the core data representation (the parameters of Gaussian primitives), making it difficult to remove or tamper with.

**(3) Facilitated detection**: Even without rendering, watermarks can be detected from suspect 3DGS models that are potentially infringing, simplifying copyright verification.

## C  DETAILED MODEL ARCHITECTURE

In Section 4.1 of the main text, we introduced the overall architecture of the watermark injector and message extractor. Here, we present a detailed description of their internal structures. As illustrated in Figure 8, we annotate each module with comprehensive specifications, including input and output details. The numbers enclosed in red parentheses indicate the corresponding dimensional information.

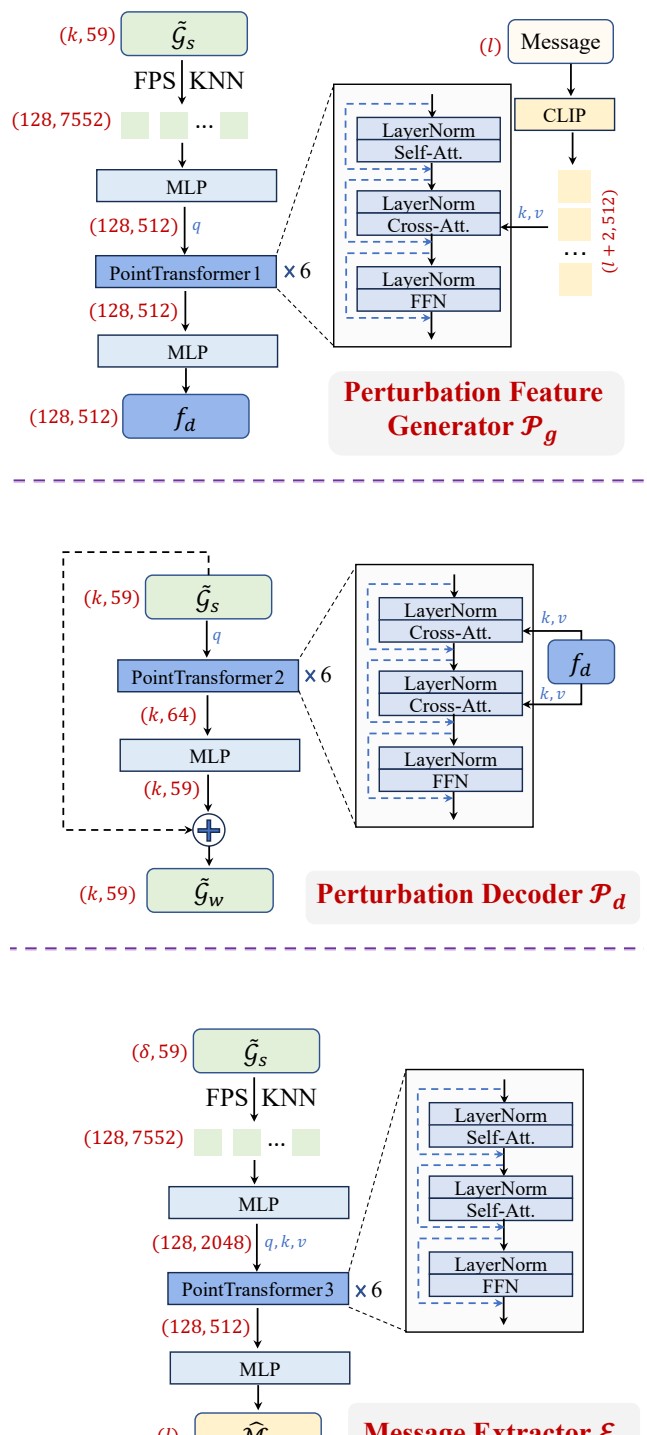

Figure 8: Schematic diagram of the detailed structure of each module in NGS-Marker.

## D    DETAILED DATASET INFORMATION

The training and testing datasets used in our study are sourced from four main origins: Blender Mildenhall et al. (2021), LLFF Mildenhall et al. (2019), Mip-NeRF 360 Barron et al. (2022), and several commonly used scenes from the 3D editing domain Haque et al. (2023). The specific scenes include:

Table 6: Experimental results for the clean collaboration baseline.

| Type | PSNR/SSIM ↑ | LPIPS ↓ | 3D-Acc ↑ | Bit-Acc ↑ |
|------|-------------|---------|----------|-----------|
| Indirect | 34.63 / 0.981 | 0.034 | N/A | 99.02[*] |
| Native | 40.17 / 0.995 | 0.007 | 97.90 | 98.35[§] |
| Hybrid | 38.16 / 0.986 | 0.020 | 97.20 | 98.27[*]
97.33[§] |

drums, ficus, hotdog, lego, materials, mic, ship, bicycle, bonsai, counter, flowers, kitchen, room, stump, treehill, fern, fortress, horns, llff_flower, llff_room, orchids, trex, train, truck, chair, garden, bear, and person-small. Among these, chair, garden, bear, and person-small are designated as testset, while the remaining scenes are used for training.

During testing, we simulate partial infringement at different scales using the testset. For the chair scene, the entire scene is treated as a plagiarized segment and embedded into other scenes. In the garden scene, the vase and table are extracted and used as plagiarized content. For the bear and person-small scenes, the main foreground objects (the bear and the person, respectively) are segmented and considered as the plagiarized portions.

The calculation details of Bit-Acc in Table 1 and Table 2 are as follows: for each baseline, we render the views of the mixed scene $\mathcal{B}_{\mathcal{A}}$, extract watermark information from these rendered images, and compare it with the ground truth. For NGS-Marker, some primitives are selected from the pirated region to serve as the anchors. These anchors and their k-nearest neighbors are then fed into the message extractor to predict the embedded watermark, and the accuracy is subsequently computed. In real-world applications, the protected regions are typically unknown, and users can identify potentially plagiarized areas directly using the method described in Section 4.3, or determine which primitives may have been misused through the 3D-Acc metric. The over 95% 3D-Acc reported in Table 2 demonstrates the practical feasibility of our method. In total, we design nine mixed scenes to evaluate, which are illustrated in Figure 1, Figure 4, Figure 10, Figure 11, Figure 12, Figure 13, and Figure 14.

## E   COOPERATING WITH HIDDEN

In Table 5, we present the results of NGS-Marker in collaboration with indirect protection methods. Here, we additionally introduce a clean baseline:

$$\mathcal{L}_{\text{cooperate}}^{\text{clean}} = \mathcal{L}_w + 0.1 \cdot \text{BCE}(\mathcal{H}(\mathcal{R}(\mathcal{G}^w, \theta)), \mathcal{M}^{id}), \tag{8}$$

where $\mathcal{H}$ denotes the pretrained HiDDeN watermark decoder. Following standard protocol, 1/8 of the camera views are used for testing, while the remainder serve as training data. The experimental results are shown in Table 6.

## F   FURTHER DISCUSSION ON $\tau$

The value of $\tau$ has an impact on 3D-Acc, and we provide additional ablation results in Figure 9. As can be observed, NGS-Marker exhibits a certain degree of robustness with respect to variations in $\tau$. For the sake of comparability, we set $\tau$ to 75%, which lies between random guessing (50%) and perfect accuracy (100%) in our experiments.

## G   ADDITIONAL VISUALIZATION EXAMPLES

In Figure 4 of the main text, we presented several representative cases of our simulated partial infringement scenarios. Here, we provide additional examples for reference, as illustrated in Figure 10, Figure 11, Figure 12, Figure 13, and Figure 14.

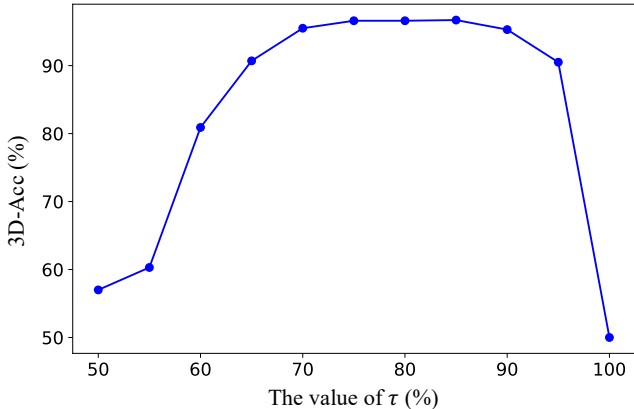

Figure 9: Ablation study of $\tau$.

Table 7: Comparison of the time required to embed watermarks by different methods.

| Time (min) | person | chair | bear | garden | Avg. |
|---|---|---|---|---|---|
| 3D-GSW | 12.8 | 9.5 | 21.4 | 27.7 | 17.9 |
| GaussianMarker | 4.2 | 4.3 | 12.7 | 16.5 | 9.4 |
| GuardSplat | 5.5 | 4.9 | 15.2 | 19.5 | 11.3 |
| NGS-Marker | 4.0 | 9.3 | 28.7 | 35.2 | 19.3 |
| NGS-Marker-Parallel | 2.7 | 4.6 | 12.5 | 13.9 | 8.4 |

Furthermore, we also evaluate the performance when objects are removed or rotated in the scenes, as illustrated in Figure 15 and Figure 16.

Moreover, we further investigate the detection results when watermarked primitives and non-watermarked primitives overlap. We place the watermarked `bear` into the `bicycle` scene and gradually increase the overlap between the bear and the ground; the visualization results are shown in Figure 17. As can be seen, our method consistently captures all visually perceivable plagiarized regions. For regions that are not visible in the rendered image, plagiarism becomes meaningless, and we therefore do not pursue further analysis.

## H USE OF LLMs

In our manuscript, we employed the LLM to perform grammatical checks on the written content.

## I TIME COMPARISON

We further report a comparison between our method and existing approaches in terms of the time required to embed watermarks across different scenes, as summarized in Table 7. Benefiting from a native watermarking scheme that operates directly on Gaussian primitives, our method can partition large scenes into subregions during watermark embedding, thereby enabling efficient parallel processing. By contrast, existing 3DGS watermarking approaches decode the watermark from rendered images, where the sets of primitives influencing different rendered views may overlap, making parallel computation difficult. In Table 7, NGS-Marker-Parallel denotes the time cost of watermark embedding when using four GPUs in parallel. Specifically, we uniformly divide each 3D scene into four parts along the x–y plane, embed the watermark into each part in parallel on four GPUs, and then recombine the four parts followed by a second-stage optimization of the merged scene. As can be observed, the watermark embedding time of all methods generally increases with scene complexity, whereas our method, when coupled with the parallelization strategy, achieves the lowest average time cost.

Table 8: Comparison of the time required to extract watermarks by different methods.

| Time (ms) | person | chair | bear | garden | Avg. |
|---|---|---|---|---|---|
| 3D-GSW | 16.2 | 17.2 | 18.7 | 19.1 | 17.8 |
| GaussianMarker | 15.0 | 16.1 | 17.5 | 18.0 | 16.7 |
| GuardSplat | 17.2 | 18.3 | 19.7 | 20.2 | 18.9 |
| NGS-Marker | 20.7 | 20.7 | 20.9 | 21.0 | 20.8 |

Table 9: Comparison of the computational complexity.

| Method | Params(M) | FLOPs (G) |
|---|---|---|
| 3D-GSW | 0.29 | 14.48 |
| GaussianMarker | 0.29 | 14.48 |
| GuardSplat | 151.48 | 4.37 |
| NGS-Marker | 51.28 | 6.61 |

In addition, we evaluate the time required for a single watermark detection. For existing 3DGS watermarking strategies, watermark detection involves rendering images followed by information extraction, whereas for our approach, detection consists of extracting a subset of Gaussian primitives and then performing information extraction. We measure the time cost of a single detection pass for different methods on a single A100 GPU, as reported in Table 8. The results show that the time required by our method for a single watermark extraction is comparable to that of existing methods, while the additional time incurred by our method grows more slowly as scene complexity increases.

## J   COMPUTATIONAL COMPLEXITY

Although we adopt a Point Transformer, the computational cost of watermark embedding and extraction does not increase sharply, mainly for two reasons: (1) our method does not require feeding the entire scene into the Point Transformer; instead, we only process a fixed number of primitives at a time; (2) before feeding the Gaussian primitives into the Point Transformer, we use FPS and KNN to convert them into a small number of tokens, which significantly reduces the attention cost inside the transformer.

To more intuitively compare the computational complexity of our method with existing approaches, we measured both the number of parameters and the computational cost for a single forward pass. For other methods that extract watermarks from images, we fixed the input image size to $224 * 224$. For our method, we fixed the number of Gaussian primitives to 8192. We used the open-source tools *thop* and *fvcore* to measure the parameter count and computation, respectively, and the results are summarized in table 9. As can be seen, CNN-based methods (3D-GSW, GaussianMarker) have relatively few parameters but a higher computational cost, whereas transformer-based methods (GuardSplat, NGS-Marker) have more parameters but a lower computational cost. Compared with these existing methods, the computational complexity of our approach does not increase significantly.

## K   MORE ABLATION STUDIES

### K.1   THE EFFECT OF $\delta$ ON ROBUSTNESS

Here, we investigate the impact of $\delta$ on watermark robustness. We vary the value of $\delta$ and measure the Bit-Acc of watermarked scenes under different types of perturbations, as reported in Table 10. Overall, we observe that decreasing $\delta$ leads to a slight reduction in robustness, but the effect remains limited as long as $\delta$ stays within a reasonable range.

When a scene is perturbed, $\delta$ also affects the number of corrupted primitives that are passed to the information extractor. A smaller $\delta$ makes the detection system appear more fragile; however, it also reduces the amount of perturbation each detection is exposed to. Consequently, for scenes subjected

Table 10: Ablation study on the effect of $\delta$ on robustness.

| $\delta$ | None | Gaussian Noise ($\sigma$=0.015) | Rotation ($\pm\pi$) | Scaling ($\pm\infty$) | Densification (0%-50%) | Dropout (0%-50%) | Translation ($\pm\infty$) |
|---|---|---|---|---|---|---|---|
| 2048 | 95.62 | 93.89 | 95.62 | 95.62 | 94.14 | 93.78 | 95.62 |
| 4096 | 96.77 | 95.19 | 96.77 | 96.77 | 95.83 | 94.94 | 96.77 |
| 6144 | 96.83 | 95.30 | 96.83 | 96.83 | 96.22 | 95.36 | 96.83 |
| 8192 | 98.35 | 97.06 | 98.35 | 98.35 | 97.93 | 97.41 | 98.35 |
| 12288 | 98.17 | 97.02 | 98.17 | 98.17 | 97.82 | 97.26 | 98.17 |

Table 11: The influence of using different attributes on performance.

| Attributes | Bit-Acc | PSNR ↑ | SSIM ↑ | LPIPS ↓ |
|---|---|---|---|---|
| position+SH(0) | 81.45 | 31.39 | 0.968 | 0.035 |
| position+SH(3) | 92.66 | 38.50 | 0.991 | 0.017 |
| position+SH(3)+opacity | 96.23 | 39.74 | 0.994 | 0.010 |
| All | 98.35 | 40.17 | 0.995 | 0.007 |

to the same level of perturbation, the robustness of the watermark does not differ substantially across different values of $\delta$.

## K.2 THE INFLUENCE OF USING DIFFERENT ATTRIBUTES ON PERFORMANCE

In our watermark embedding and extraction pipeline, we utilize all available attributes of the Gaussian primitives. To investigate how different attribute subsets affect performance, we conducted ablation studies, as shown in table 11, where SH (0) denotes 0-order SH coefficients and SH (3) denotes all SH coefficients. We observe that, in general, **the lower the total dimensionality of the used attributes, the harder it becomes to strike a good balance between watermark accuracy and rendering quality**. This empirically supports our design choice of using all available Gaussian attributes for robust and imperceptible watermarking.

## L ROBUSTNESS AGAINST ADVERSARIAL ATTACKS AND 3D EDITING

To more comprehensively investigate the properties of NGS-Marker, we examine its robustness in the presence of adversarial attacks and 3D editing operations.

### L.1 ADVERSARIAL ATTACK

Since existing 3DGS watermarking works do not consider adversarial attacks, we designed a dedicated adversarial attack pipeline tailored to 3DGS watermarking methods. Following Kerckhoffs' principle and the C&W (Carlini & Wagner) attack Carlini & Wagner (2017), we assume that the attacker knows all details of the watermarking algorithm but does not know the hyperparameters used during watermark embedding (e.g., '*bit_len*' and $\delta$), and that the message extractor remains private. In the attack process, the attacker first trains a message extractor, then sets the target message to 0.5, and finally uses gradient descent to optimize the attacked scene so that the decoded message approaches the target message:

$$\mathcal{L}_{\text{attack}} = \text{BCE}(\mathcal{M}_{\text{attack}}, \mathcal{M}_{\text{target}}) + \text{MSE}(\mathcal{R}(\mathcal{G}^w), \mathcal{R}(\mathcal{G}^s)), \tag{9}$$

where, $\mathcal{M}_{\text{attack}}$ denotes the watermark message extracted from the attacked scene by the attacker's extractor, and $\mathcal{M}_{\text{target}}$ is a target message sequence whose entries are all 0.5. During the attack process, for the baseline method, the attack gradients are propagated through the rendered images to the Gaussian primitives, resulting in an indirect attack; in contrast, in our method, the attack gradients are propagated directly to the Gaussian primitives, enabling a direct attack. For the HiDDeN extractor used in GaussianMarker and 3D-GSW, we train the extractor on 10,000 images sampled from COCO; for GuardSplat, we retrain the extractor using the algorithms in their public codebase. For fair comparison, we fix the number of attack iterations to 300 for all methods. We conduct

Table 12: Robustness against adversarial attacks.

| Attacker | 8 bits $\delta$:8192 | 16 bits $\delta$:4096 | 16 bits $\delta$:8192 | 24 bits $\delta$:8192 | Avg. |
|---|---|---|---|---|---|
| 3D-GSW | 57.29 | 58.55 | 58.55 | 67.19 | 60.40 |
| GaussianMarker | 57.74 | 57.39 | 57.39 | 63.30 | 59.00 |
| GuardSplat | 62.03 | 65.42 | 65.42 | 72.51 | 66.35 |
| NGS-Marker | 93.50 | 79.22 | 65.83 | 92.16 | 82.68 |

Table 13: Robustness against 3D editing.

| Bit-Acc | *"Turn the bear into a grizzly bear"* | *"Turn him into a clown"* |
|---|---|---|
| 3D-GSW | 72.40 | 74.39 |
| GaussianMarker | 70.26 | 69.42 |
| GuardSplat | 78.55 | 77.18 |
| NGS-Marker | 91.06 | 90.53 |

attacks on the `bear` scene embedded with a 16-bit message ($\delta$=8192), and the experimental results of Bit-Acc are shown in table 12.

We observe that NGS-Marker and the baseline methods exhibit clearly different behaviors under adversarial attacks. When the attacker does not correctly guess the hyperparameters we use during watermark embedding, our method is able to effectively withstand the attack; only when the hyperparameters are exactly matched does the watermark accuracy of our method drop significantly. In contrast, for existing 3DGS watermarking methods that rely on rendered images for indirect protection, different hyperparameter choices already cause severe degradation of the embedded watermark.

### L.2 3D Editing

We utilize GaussianEditor Chen et al. (2024b) to edit the watermarked scenes and then detect the watermark information in the edited scenes. Specifically, we select two scenes, `bear` and `person`, and apply the editing prompts *"Turn the bear into a grizzly bear"* and *"Turn him into a clown,"* respectively. The Bit-Acc results are reported in Table 13. As can be seen, our method is affected the least even under 3D edits that significantly alter object appearance, whereas the accuracy of existing indirect watermarking approaches drops substantially.

## M Experimental Results on Complex and Dynamic Scenes

In this section, we further present the experimental results of NGS-Marker on the complex scenarios in the T&T dataset Knapitsch et al. (2017) and the dynamic scenes in the D-NeRF dataset Pumarola et al. (2021).

### M.1 Complex Scenes

The results of our experiments on four complex scenarios (`train`, `truck`, `drjohnson`, `playroom`) are presented in Table 14, from which we observe that the accuracy of our method in these scenarios is comparable to that of the original evaluation.

### M.2 Dynamic Scenes

To represent dynamic scenes, we adopt the classical 4DGS method Wu et al. (2024b). In 4DGS, the primitives at any time step $t$ are not explicitly stored but are predicted by the *deform_network*. Therefore, during watermark embedding we also optimize the weights of the *deform_network*, with the objective that any local region of the decoded 3D scene at any time step should contain the

Table 14: Experimental results on complex scenes.

| Scene | 8 bits | | 16 bits | | 24 bits | |
|---|---|---|---|---|---|---|
| | Bit-Acc | PSNR ↑ | Bit-Acc | PSNR ↑ | Bit-Acc | PSNR ↑ |
| Train | 98.75 | 41.07 | 97.83 | 40.12 | 94.50 | 39.05 |
| Truck | 99.03 | 41.00 | 97.84 | 39.61 | 94.73 | 39.20 |
| Drjohnson | 98.06 | 39.48 | 96.46 | 39.13 | 93.22 | 38.43 |
| Playroom | 98.32 | 40.19 | 97.59 | 39.38 | 93.65 | 38.71 |
| Original Test Results | 99.26 | 41.77 | 98.35 | 40.17 | 94.79 | 39.61 |

Table 15: Experimental results on dynamic scenes. The numbers in parentheses denote the results on static scenes under the corresponding settings.

| | Bit-Acc | PSNR ↑ | SSIM ↑ | LPIPS ↓ |
|---|---|---|---|---|
| 8 bits | 98.71 (99.26) | 41.50 (41.77) | 0.996 (0.996) | 0.004 (0.004) |
| 16 bits | 98.24 (98.35) | 40.29 (40.17) | 0.995 (0.995) | 0.008 (0.007) |
| 24 bits | 95.22 (94.79) | 39.48 (39.61) | 0.993 (0.993) | 0.013 (0.013) |

complete watermark information:

$$\mathcal{L}_{\text{dynamic}} = \text{BCE}(\mathcal{M}^{\text{id}}, \mathcal{E}(\text{Sample}(\mathcal{D}(\mathcal{G}^w, t)))) + \text{MSE}(\mathcal{R}(\mathcal{D}(\mathcal{G}^w, t)), \mathcal{R}(\mathcal{D}(\mathcal{G}^s, t))), \quad (10)$$

where, $\mathcal{E}$ denotes the message extractor, Sample denotes the local Gaussian primitive sampling operation, D denotes the *deform_network*, and $t$ is the time step in the dynamic scene. We conduct experiments on the D-NeRF dataset, and the results are shown in table 15 (for ease of comparison, we also list in parentheses the corresponding results in static scenes). As can be seen, our method can be almost seamlessly extended to dynamic scenes.

## N    WHY CAN GAUSSIAN PRIMITIVES BE USED FOR WATERMARK EMBEDDING

In Sec. 3, we use a simple experiment to demonstrate the feasibility of employing neural networks to embed invisible watermarks into unstructured noise. Here, we provide a more theoretical explanation of why neural networks are capable of injecting imperceptible watermark information into Gaussian primitives.

**(1) High-dimensional explicit parameters & over-parameterization.** Each Gaussian $\mathcal{G}_i = (\mu_i, \alpha_i, s_i, c_i, r_i)$ has multiple continuous degrees of freedom (position, scale, opacity, SH coefficients, rotation, etc.), so a local set of Gaussian primitives forms an extremely high-dimensional explicit vector, while during rendering it only affects a limited number of pixels. Under a differentiable renderer $\mathcal{R}$, in the first-order approximation $\mathcal{R}(\mathcal{G} + \Delta\mathcal{G}) \approx \mathcal{R}(\mathcal{G}) + J\Delta\mathcal{G}$, the number of pixels is much smaller than the parameter dimensionality, so the Jacobian $J$ has a large "*approximate null space*." Thus, there exist many directions of $\Delta\mathcal{G}$ that leave the rendered image almost unchanged while significantly changing the Gaussian parameters, which is precisely the capacity for embedding imperceptible watermarks. The MSE term in the loss encourages $\Delta\mathcal{G}$ to lie in these low-sensitivity directions.

**(2) Spatial structure & learnability.** Unlike pure noise, 3D Gaussians are concentrated near object surfaces, and quantities such as centers, scales, and SH coefficients vary smoothly and are correlated within local neighborhoods. This spatial structure allows networks such as PointTransformer to exploit these regularities and learn an invertible embedding/decoding mapping.

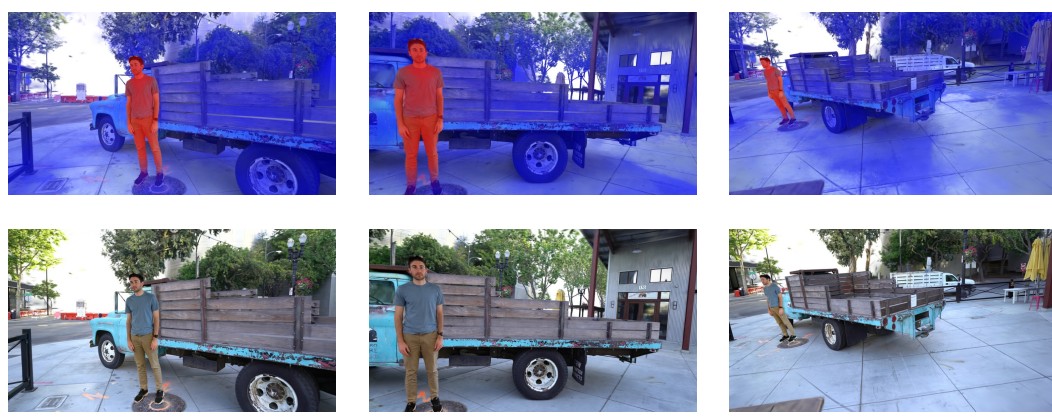

Figure 10: Additional visualization examples.

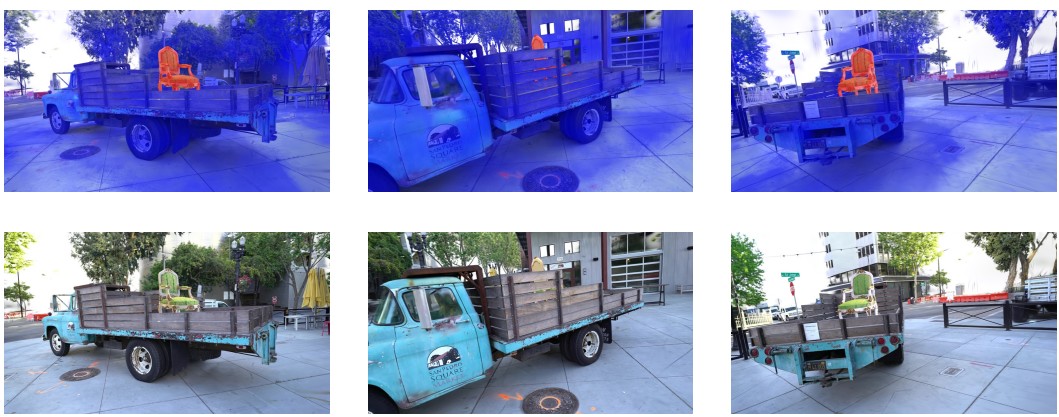

Figure 11: Additional visualization examples.

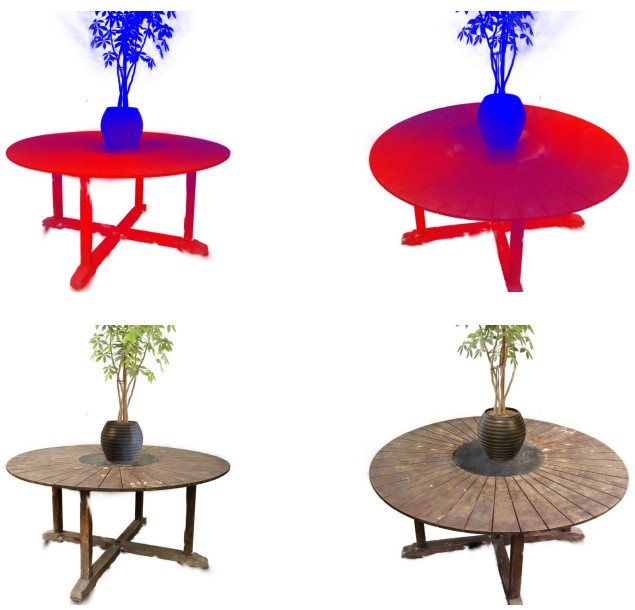

Figure 12: Additional visualization examples.

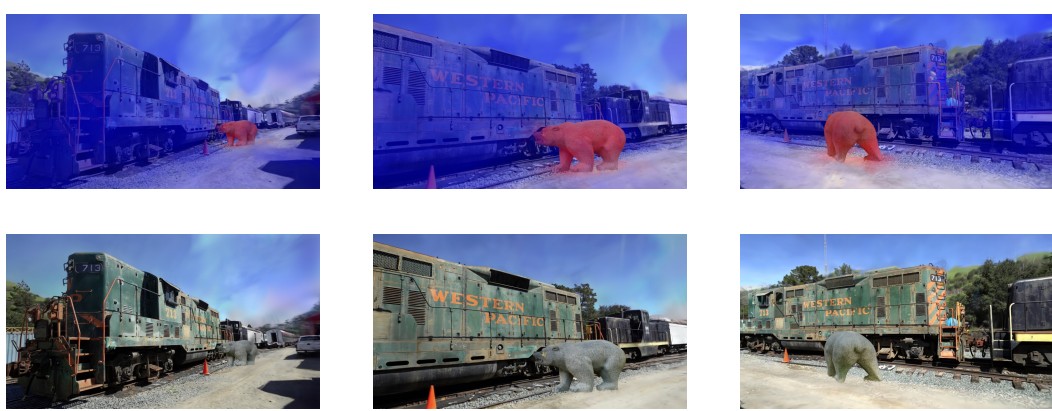

Figure 13: Additional visualization examples.

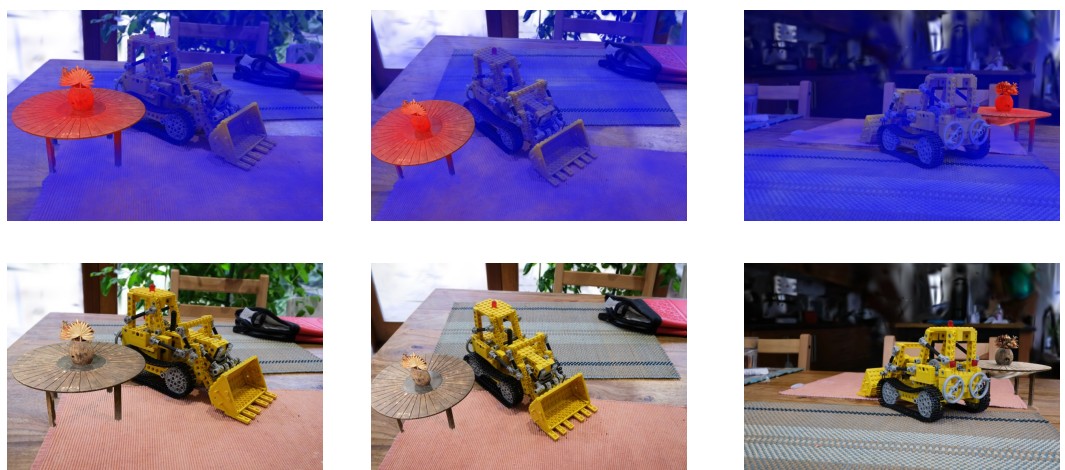

Figure 14: Additional visualization examples.

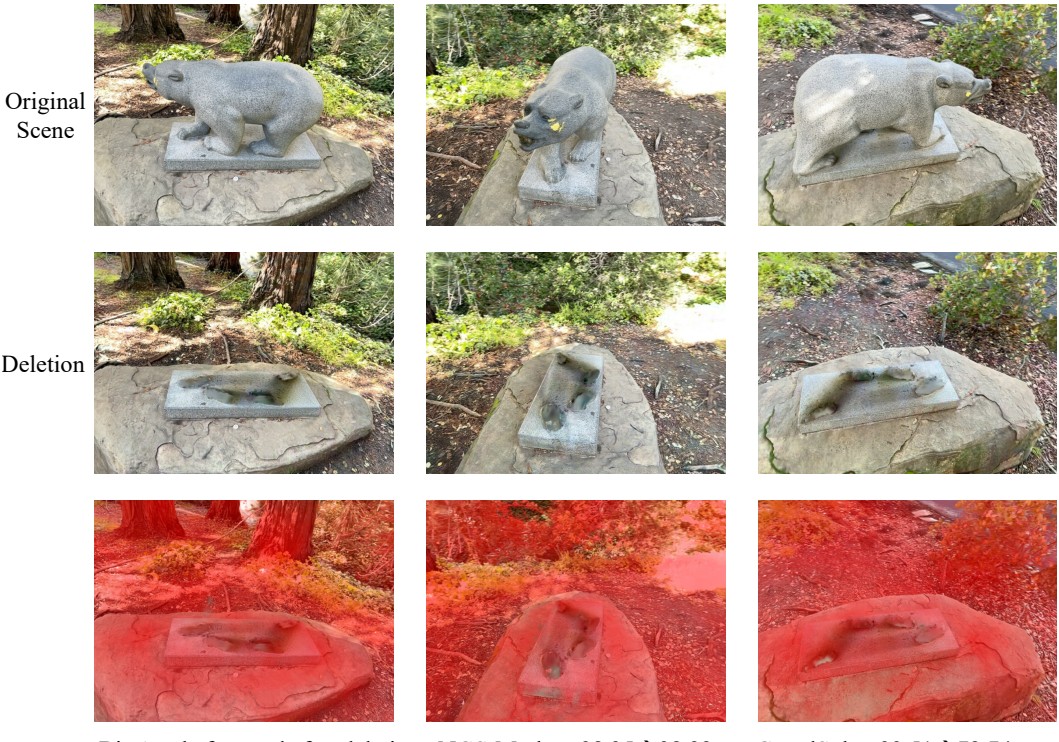

Bit-Acc before and after deletion:  NGS-Marker: 98.35→98.22    GuardSplat: 99.51→72.74

Figure 15: Deletion experiment.

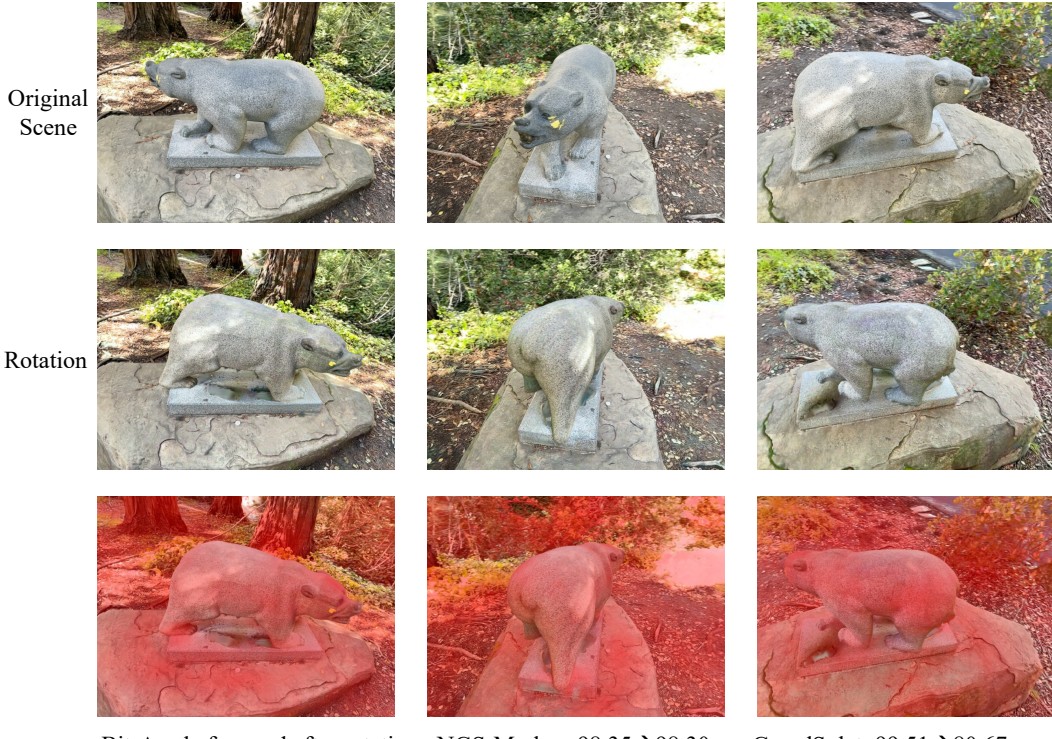

Bit-Acc before and after rotation:  NGS-Marker: 98.35→98.30    GuardSplat: 99.51→80.67

Figure 16: Rotation experiment.

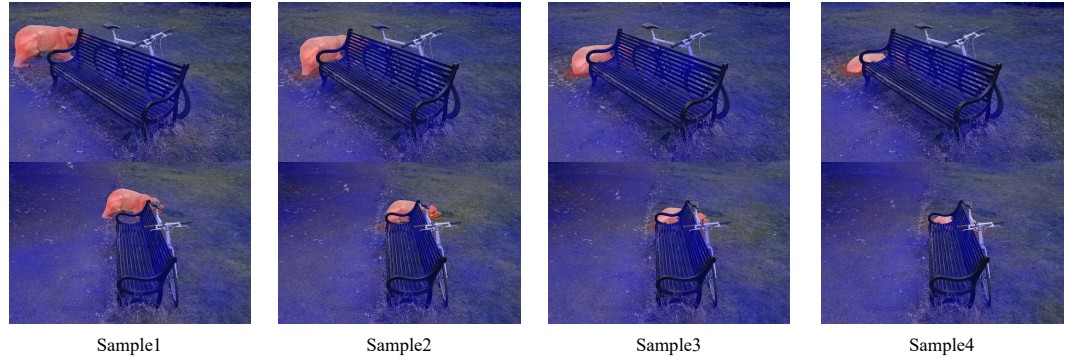

Sample1        Sample2        Sample3        Sample4

Figure 17: Visualization results of overlapping Gaussian primitives with and without watermarking.

