# OpenReview forum: "NGS-Marker: Robust Native Watermarking for 3D Gaussian Splatting"
_ICLR.cc/2026/Conference — ICLR 2026 Poster_

### Official Review · Reviewer_5NBU · 2025-10-29

**Soundness:** 3
**Presentation:** 3
**Contribution:** 3
**Rating:** 6
**Confidence:** 5

**Summary:**

This paper proposes NGS-Marker, a native watermarking framework for 3D Gaussian Splatting (3DGS) that addresses the problem of partial infringement. Unlike existing indirect methods that watermark rendered images, NGS-Marker embeds watermarks directly into 3D Gaussian primitives, enabling detection even when only subsets of primitives are extracted and reused. The method uses a jointly trained watermark injector and message extractor with a progressive optimization strategy.

**Strengths:**

## Strengths

1. The paper identifies a critical and underexplored vulnerability in 3DGS assets - partial infringement, where adversaries extract and reuse subsets of Gaussian primitives. The motivation is clearly articulated with concrete examples.

2. The method achieves native watermarking embedding and extraction.

3. The hybrid protection mechanism and personalized watermarking demonstrate practical flexibility.

**Weaknesses:**

## Weaknesses

1. While the feasibility analysis (Section 3, Figure 2) shows that HiDDeN can embed watermarks in random noise, the theoretical connection to why this should work for structured 3D Gaussian primitives is weak. The paper lacks analysis of what properties of Gaussian primitives make them suitable for watermark embedding.

2. Table 4 shows embedding time increases significantly with scene complexity (4 min for 42K primitives to 35 min for 589K primitives)
   The authors acknowledge in limitations that "designing an efficient partitioning and traversal strategy could further improve scalability" but don't provide solutions.
   How does the method scale to production-level scenes with millions of primitives?

3. No analysis of adversarial attacks specifically designed to remove native watermarks
   What if an adversary fine-tunes or re-optimizes the Gaussian primitives?
   The robustness experiments (Table 3) only test standard distortions, not adversarial removal attempts.

4. Although the method achieves native embedding, the method is built on top of the PointTransformer. What parameters are used in the injection and extraction? Are all parameters used for injection and extraction? There should be an ablation study for different parameters that are selected for injection and decoding.
For example, if we can only use the color and position attributes, can we extend the method to become compatible with point cloud data?

5.  The partial infringement simulation may not fully capture real-world misuse scenarios. More challenging infringements should be discussed such as 3D adversarial attacks, Semantic editing[1] or recoloring[2]

[1] GaussianEditor: Swift and Controllable 3D Editing with Gaussian Splatting

[2] GeometrySticker: GeometrySticker: Enabling Ownership Claim of Recolorized Neural Radiance Fields

**Questions:**

## Questions

1. What is the false positive rate? How often does the method incorrectly identify non-watermarked primitives as watermarked?

2. How does the method achieve indirect message encoding and decoding?

Do you optimize the injected with the pretrained frozen message decoder just like the approaches in [1][2]?

How different is the optimization process for the indirect, native, and hybrid optimization?

Do you first optimize for native and then for indirection protection? Or do you optimize them simultaneously?

I think the explanation for this port should be clearer.

3. The paper focuses on static 3DGS, but many applications involve dynamic content.

Can the proposed method be extended to dynamic scenes?

4.  Only 4 test scenes are quite limited for drawing strong conclusions.
     Are the injector and extractor pretrained on the 24 training scenes and then only validated on the 4 test scenes?
    How are the validation results on the training datasets?
     Beyond the 4 test scenes, how does the method perform on more complex, diverse scenes such as Tanks and Temples Benchmark?

[1] WateRF: Robust Watermarks in Radiance Fields for Protection of Copyrights

[2] 3D-GSW: 3D Gaussian Splatting for Robust Watermarking

---

> ### Author Response · Authors · 2025-11-26
> **Response to W1 and W2**
>
> **W1. While the feasibility analysis (Section 3, Figure 2) shows that HiDDeN can embed watermarks in random noise, the theoretical connection to why this should work for structured 3D Gaussian primitives is weak. The paper lacks analysis of what properties of Gaussian primitives make them suitable for watermark embedding.**
>
> **Response to W1:** We thank you for pointing out that the intuition underlying Figure 2 was under-explained in the original version. Our goal in Section 3 is to use HiDDeN-on-noise as a sanity check showing that a neural encoder–decoder can reliably embed and extract messages from completely unstructured inputs, and to thereby make a preliminary assessment of whether a similarly non–explicitly-structured representation such as 3DGS can also support watermark embedding and extraction via a similar codec architecture. We agree that our explanation of why Gaussian primitives are suitable carriers for watermarks can be made more explicit. In the revised version, we have added a more detailed theoretical analysis in App. N, and we also briefly summarize and reference this analysis in the main text (line 187). Specifically, we analyze, based on the properties of 3D Gaussian Splatting (3DGS), why Gaussian primitives can be used to embed watermarks:
>
> > **High-dimensional explicit parameters & over-parameterization.** Each Gaussian $\mathcal{G} _ i={(\mu _ I, \alpha _ I, s _ I, c _ I, r _ i)}$ has multiple continuous degrees of freedom (position, scale, opacity, SH coefficients, rotation, etc.), so a local set of Gaussian primitives forms an extremely high-dimensional explicit vector, while during rendering it only affects a limited number of pixels. Under a differentiable renderer $\mathcal{R}$, in the first-order approximation $\mathcal{R}(\mathcal{G} + \Delta \mathcal{G}) \approx \mathcal{R}(\mathcal{G}) + J \Delta \mathcal{G}$, the number of pixels is much smaller than the parameter dimensionality, so the Jacobian $J$ has a large "*approximate null space.*" Thus, there exist many directions of $\Delta G$ that leave the rendered image almost unchanged while significantly changing the Gaussian parameters, which is precisely the capacity for embedding imperceptible watermarks. The MSE term in the loss encourages $\Delta G$ to lie in these low-sensitivity directions.
>
> > **Spatial structure & learnability.** Unlike pure noise, 3D Gaussians are concentrated near object surfaces, and quantities such as centers, scales, and SH coefficients vary smoothly and are correlated within local neighborhoods. This spatial structure allows networks such as PointTransformer to exploit these regularities and learn an invertible embedding/decoding mapping.
>
> ---
>
> **W2. The authors acknowledge in limitations that "designing an efficient partitioning and traversal strategy could further improve scalability" but don't provide solutions. How does the method scale to production-level scenes with millions of primitives?**
>
> **Response to W2:** In the revised version of the paper, we have added to App. I a simple yet practical parallel strategy to accelerate watermark embedding, together with corresponding experiments. Specifically, we uniformly partition the 3D scene into four regions on the x–y plane, use four GPUs to embed watermarks into each region in parallel, and then recombine the four regions and perform a second-stage optimization on the merged scene. The comparison of watermark embedding time is reported in the table below, from which we can see that the parallel scheme substantially reduces the time required for watermark injection. It is also reasonable to expect that, for scenes containing millions of primitives, as long as a sufficient number of GPUs is available to support block-wise watermarking, the embedding time can be kept at a relatively low level.
>
> Time (min) | Person | Chair | Bear | Garden | Avg.
> --- | :---: | :---: | :---: | :---: | :---:
> NGS-Marker | 4.0 | 9.3 | 28.7 | 35.2 | 19.3
> NGS-Marker-Parallel | 2.7 | 4.6 | 12.5 | 13.9 | 8.4
>
> The feasibility of this parallel acceleration essentially stems from our proposed "native watermark" design. Existing 3DGS watermarking methods rely on rendered images as an intermediate medium during both embedding and detection; however, different images may be associated with the same Gaussian primitive, which makes it difficult for these methods to support efficient parallelization. In contrast, our method operates directly on the Gaussian primitives, and therefore naturally fits parallel optimization strategies.

---

> ### Author Response · Authors · 2025-11-26
> **Response to W3 and W5**
>
> **W3. No analysis of adversarial attacks specifically designed to remove native watermarks.**
>
> **Response to W3:** Thank you for reviewing our manuscript. Since existing 3DGS watermarking works do not consider adversarial attacks, we designed a dedicated adversarial attack pipeline tailored to 3DGS watermarking methods. Following Kerckhoffs’ principle and the C&W (Carlini & Wagner) attack [1], we assume that the attacker knows all details of the watermarking algorithm but does not know the hyperparameters used during watermark embedding (e.g., `bit_len` and $\delta$), and that the message extractor remains private. In the attack process, the attacker first trains a message extractor, then sets the target message to 0.5, and finally uses gradient descent to optimize the watermarked scene so that the decoded message approaches the target message:
>
> $\mathcal{L} _ {\text{attack}}=\mathrm{BCE}(\mathcal{M} _ \text{attack}, \mathcal{M} _ {\text{target}}) + \mathrm{MSE}(\mathcal{R}(\mathcal{G} ^ {w}), \mathcal{R}(\mathcal{G} ^ {s})),$
>
> where, $\mathcal{M} _ \text{attack}$ denotes the watermark message extracted from the attacked scene by the attacker’s extractor, and $\mathcal{M} _ \text{target}$ is a target message sequence whose entries are all 0.5. During the attack process, for the baseline method, the attack gradients are propagated through the rendered images to the Gaussian primitives, resulting in an indirect attack; in contrast, in our method, the attack gradients are propagated directly to the Gaussian primitives, enabling a direct attack. For the HiDDeN extractor used in GaussianMarker and 3D-GSW, we train the extractor on 10,000 images sampled from COCO; for GuardSplat, we retrain the extractor using the algorithms in their public codebase. For fair comparison, we fix the number of attack iterations to 300 for all methods. We conduct attacks on the "*bear*" scene embedded with a 16-bit message ($\delta$ = 8192), and the experimental results of Bit-Acc are shown in the table below.
>
> | Attacker     | 8 bits $\delta$:8192 | 16 bits $\delta$:4096 | 16 bits $\delta$:8192 | 24 bits $\delta$:8192 | Avg.  |
> |----------------|:---------------:|:----------------:|:----------------:|:----------------:|:-------:|
> | 3D-GSW         | 57.29         | 58.55          | 58.55          | 67.19          | 60.40 |
> | GaussianMarker | 57.74         | 57.39          | 57.39          | 63.30          | 59.00 |
> | GuardSplat     | 62.03         | 65.42          | 65.42          | 72.51          | 66.35 |
> | NGS-Marker     | 93.50         | 79.22          | 65.83          | 92.16          | 82.68 |
>
> We observe that NGS-Marker and the baseline methods exhibit clearly different behaviors under adversarial attacks. When the attacker does not correctly guess the hyperparameters we use during watermark embedding, our method is able to effectively withstand the attack; only when the hyperparameters are exactly matched does the watermark accuracy of our method drop significantly. In contrast, for existing 3DGS watermarking methods that rely on rendered images for indirect protection, different hyperparameter choices already cause severe degradation of the embedded watermark. We have added this set of experiments to App. L.
>
> [1] Carlini N, Wagner D. Towards evaluating the robustness of neural networks[C]//2017 ieee symposium on security and privacy (sp). Ieee, 2017: 39-57.
>
> ---
>
> **W5. More challenging infringements should be discussed such as 3D adversarial attacks, Semantic editing or recoloring.**
>
> **Response to W5:** Thank you very much for your valuable suggestion. Following your comment, we have added adversarial attack experiments in **Response to W3**. In addition, we now report experiments for two types of 3D editing (semantic changes and color changes). Specifically, we select two scenes, "*bear*" and "*person," for evaluation. Using GaussianEditor [2], we edit these scenes with the prompts "*Turn the bear into a grizzly bear*" and "*Turn him into a clown,*" respectively, and then measure the watermark accuracy after editing. The Bit-Acc results are summarized in the table below. As shown, our method is least affected by 3D edits that significantly alter the appearance, whereas the accuracy of existing indirect watermarking methods drops substantially.
>
> | Bit-Acc       | *"Turn the bear into a grizzly bear"* | *"Turn him into a clown"* |
> |---------------|:-----------------:|:------------------------:|
> | 3D-GSW        | 72.40                             | 74.39                  |
> | GaussianMarker| 70.26                             | 69.42                  |
> | GuardSplat    | 78.55                             | 77.18                  |
> | NGS-Marker    | 91.06                             | 90.53                  |
>
> [2] Chen Y, Chen Z, Zhang C, et al. Gaussianeditor: Swift and controllable 3d editing with gaussian splatting[C]//Proceedings of the IEEE/CVF conference on computer vision and pattern recognition. 2024: 21476-21485

---

> ### Author Response · Authors · 2025-11-26
> **Response to W4, Q1, and Q2**
>
> **W4. What parameters are used in the injection and extraction? Are all parameters used for injection and extraction? There should be an ablation study for different parameters that are selected for injection and decoding. For example, if we can only use the color and position attributes, can we extend the method to become compatible with point cloud data?**
>
> **Response to W4:** We sincerely thank you for the insightful comments. In our watermark embedding and extraction pipeline, **we utilize all available attributes of the Gaussian primitives**. To investigate how different attribute subsets affect performance, we conducted ablation studies, as shown in the table below, where SH (0) denotes 0-order SH coefficients and SH (3) denotes all SH coefficients. We observe that, in general, **the lower the total dimensionality of the used attributes, the harder it becomes to strike a good balance between watermark accuracy and rendering quality**. This empirically supports our design choice of using all available Gaussian attributes for robust and imperceptible watermarking.
>
> | Attributes | Bit-Acc | PSNR↑ | SSIM↑ | LPIPS↓ |
> | - | :-: | :-: | :-: | :-: |
> | position+SH (0) |   81.45 | 31.39 | 0.968 |  0.035 |
> | position+SH (3)|   92.66 | 38.50 | 0.991 |  0.017 |
> | position+SH (3) +opacity |   96.23 | 39.74 | 0.994 |  0.010 |
> | All | 98.35 | 40.17 | 0.995 |  0.007 |
>
> In principle, *our method can also be applied to point-cloud data, but its effectiveness is reduced compared with 3DGS*. In the table, the configuration "position + SH (0)" is the closest to a point-cloud-like setting, and we can see that both watermark accuracy and rendering quality degrade noticeably under this configuration. We hypothesize that this is because, in 3D Gaussian Splatting, different attributes (scale, opacity, rotation, SH coefficient, position.) are naturally coupled and provide a rich, redundant parameter space: the network can jointly adjust multiple attributes in a coordinated manner to hide information while keeping the rendered appearance nearly unchanged. In contrast, a pure point-cloud–like representation with only three position coordinates and three color channels offers much less redundancy; any modification to these few attributes more directly alters geometry or appearance, making it more difficult to embed information without visible distortion. We appreciate your suggestion, and we have added the corresponding experiments and discussion to App. K in the revised manuscript.
>
> ---
>
> **Q1. What is the false positive rate? How often does the method incorrectly identify non-watermarked primitives as watermarked?**
>
> **Response to Q1:** In Table 2 of the manuscript, when computing 3D-Acc, we sampled equal numbers of Gaussian primitives from both watermarked and non-watermarked regions as anchor points. Here, we additionally report the probability that the model predicts a primitive to be watermarked when the anchor points are sampled only from the non-watermarked region. As shown in the table below, the false positive rate of our method is around 3%. We speculate that most of these misclassified primitives come from those near the boundaries between watermarked and non-watermarked regions.
>
> | | 8 bits |16 bits|24 bits|
> |-| :-: | :-: | :-: |
> |FPR (%)|3.7|2.8|3.1|
>
> ---
>
> **Q2. How does the method achieve indirect message encoding and decoding?**
>
> **Q2.1** Do you optimize the injected with the pretrained frozen message decoder just like the approaches in WateRF and 3D-GSW?
>
> **Response to Q2.1:** Yes. For the data in Table 5, we used the same method as the baseline[3][4][5] when implementing the indirect optimization.
>
> **Q2.2** How different is the optimization process for the indirect, native, and hybrid optimization?
>
> **Response to Q2.2:** By indirect optimization, we mean that we adopt the optimization objective used in the baseline: the rendered images should allow the desired watermark to be decoded, while keeping the change in rendering quality small. By native optimization, we refer to the optimization objective proposed in our paper: the desired watermark should be decodable from any local Gaussian primitive, again under the constraint that the rendering quality does not change significantly. Hybrid optimization means that we use both the indirect optimization and native optimization objectives simultaneously.
>
> **Q2.3** Do you first optimize for native and then for indirection protection? Or do you optimize them simultaneously?
>
> **Response to Q2.3:** We optimize them simultaneously.
>
> **Q2.4** I think the explanation for this port should be clearer.
>
> **Response to Q2.4:** We sincerely appreciate your suggestion. We have added the relevant descriptions in Section 4.4 of the paper to make our method clearer.
>
>
> [3] 3d-gsw: 3d gaussian splatting for robust watermarking
>
> [4] Gaussianmarker: Uncertainty-aware copyright protection of 3d gaussian splatting
>
> [5] GuardSplat: Efficient and Robust Watermarking for 3D Gaussian Splatting

---

> ### Author Response · Authors · 2025-11-26
> **Response to Q3 and Q4**
>
> **Q3. The paper focuses on static 3DGS, but many applications involve dynamic content. Can the proposed method be extended to dynamic scenes?**
>
> **Response to Q3:**  In App. M, we have added the experimental results of NGS-Marker on dynamic scenes. To represent dynamic scenes, we adopt the classical 4DGS method [6]. In 4DGS, the primitives at any time step $t$ are not explicitly stored but are predicted by the `deform_network`. Therefore, during watermark embedding we also optimize the weights of the `deform_network`, with the objective that any local region of the decoded 3D scene at any time step should contain the complete watermark information:
>
> $\mathcal{L} _ {\text{dynamic}} = \mathrm{BCE}(\mathcal{M} ^ {\text{id}}, \mathcal{E}(\text{Sample}(\mathcal{D}(\mathcal{G} ^ w, t)))) + \mathrm{MSE}(\mathcal{R}(\mathcal{D}(\mathcal{G} ^ {w},t)), \mathcal{R}(\mathcal{D}(\mathcal{G} ^ {s},t))),$
>
> where, $\mathcal{E}$ denotes the message extractor, Sample denotes the local Gaussian primitive sampling operation, $\mathcal{D}$ denotes the `deform_network`, and $t$ is the time step in the dynamic scene. We conduct experiments on the D-NeRF dataset, and the results are shown in the table below (for ease of comparison, we also list in parentheses the corresponding results in static scenes). As can be seen, our method can be almost seamlessly extended to dynamic scenes.
>
> | Bits | Bit-Acc | PSNR↑  | SSIM↑  | LPIPS↓  |
> |-|:-:|:-:|:-:|:-:|
> | 8 bits | 98.71 (99.26) | 41.50 (41.77) | 0.996 (0.996) | 0.004 (0.004) |
> | 16 bits| 98.24 (98.35) | 40.29 (40.17) | 0.995 (0.995) | 0.008 (0.007) |
> | 24 bits| 95.22 (94.79) | 39.48 (39.61) | 0.993 (0.993) | 0.013 (0.013) |
>
>
> [6] Wu G, Yi T, Fang J, et al. 4d gaussian splatting for real-time dynamic scene rendering[C] // Proceedings of the IEEE/CVF conference on computer vision and pattern recognition. 2024: 20310-20320.
>
> ---
>
> **Q4. Are the injector and extractor pretrained on the 24 training scenes and then only validated on the 4 test scenes? How are the validation results on the training datasets? Beyond the 4 test scenes, how does the method perform on more complex, diverse scenes such as Tanks and Temples Benchmark?**
>
> **Response to Q4:** Yes, we train the model on 24 scenes and then tested it on 4 additional scenes that are not included in the training set. The results on the training scenes are reported below; as can be seen, the evaluation performance on the training scenes is close to that on the test scenes.
>
> | Set      | 8 bits Bit-Acc | 8 bits PSNR | 16 bits Bit-Acc | 16 bits PSNR | 24 bits Bit-Acc | 24 bits PSNR |
> | -------- | :--------------: | :-----------: | :---------------: | :------------: | :---------------: | :------------: |
> | Training | 99.31          | 41.72       | 98.28           | 40.19        | 95.30           | 39.74        |
> | Test     | 99.26          | 41.77       | 98.35           | 40.17        | 94.79           | 39.61        |
>
> In addition, we conducted experiments on four scenes in the T&T dataset ("*train*", "*truck*", "*drjohnson*", and "*playroom*"), and the results are shown in the table below. As can be observed, our method is able to produce stable watermarking performance even in complex scenes. We have added the experimental results on the T&T dataset to App. M.
>
> | Scene | 8 bits Bit-Acc | 8 bits PSNR | 16 bits Bit-Acc | 16 bits PSNR | 24 bits Bit-Acc | 24 bits PSNR |
> |-|:-:|:-:|:-:|:-:|:-:|:-:|
> | *Train* | 98.75 | 41.07       | 97.83 | 40.12        | 94.50           | 39.05        |
> | *Truck* | 99.03  | 41.00       | 97.84 | 39.61        | 94.73           | 39.20        |
> | *Drjohnson* | 98.06  | 39.48 | 96.46           | 39.13        | 93.22           | 38.43        |
> | *Playroom* | 98.32          | 40.19       | 97.59           | 39.38        | 93.65           | 38.71        |

---

### Official Review · Reviewer_NYv5 · 2025-10-30

**Soundness:** 2
**Presentation:** 3
**Contribution:** 1
**Rating:** 2
**Confidence:** 4

**Summary:**

To address Partial Infringement of 3D Gaussian Splatting (3DGS), where an adversary extracts and reuses only a subset of Gaussians, this paper presents NGS-Marker, a native watermarking framework for 3DGS by integrating a jointly trained watermark injector and message decoder and adopting a gradient-based progressive injection strategy, to achieve robust ownership decoding from any local region. The NGS-Marker is further extended with hybrid protection (combining native and indirect watermarks) while personalized watermarking is supported to embed image watermark. The experimental results show that NGS-Marker effectively defends against partial infringement while offering practical flexibility for real-world deployment.

**Strengths:**

A new watermarking framework named NGS-Maker is proposed for 3D Gaussian Splatting (3DGS) by integrating a jointly trained watermark injector and message decoder and adopting a gradient-based progressive injection strategy, to achieve robust ownership decoding from any local region. Hybrid protection (combining native and indirect watermarks) and personalized watermarking are both supported. The experimental results show the efficacy of NGS-Marker.

**Weaknesses:**

First, the concepts of indirect watermarking and native watermarking have not been clearly explained or referred, making it difficult to compare the proposed watermarking framework named NGS-Maker with the related methods.

Similarly, the definition of Partial Infringement also needs to be referred and compared with terms used in the literature.

Third, the performance comparison between the proposed method with three baselines is not enough and the performance analysis in robustness is very limited.

Last, injecting image-based watermarks into 3D scenes using NGS-Marker is okay but not very personalized.

**Questions:**

1. What mostly make the proposed watermarking framework NGS-Maker different from the existing methods?

2. What are the concrete concepts/definitions of “indirect watermarking” and “native watermarking”?

3. What are the methods in the literature most similar to the proposed one?

---

> ### Author Response · Authors · 2025-11-26
> **Response to W1, W2, and W4**
>
> **W1. The concepts of indirect watermarking and native watermarking have not been clearly explained or referred.**
>
> **Response to W1:** Thank you for pointing out that the terminology in the original submission was not sufficiently explicit. We have carefully revised the paper to clearly define *indirect watermarking* and *native watermarking*. In the revised Introduction, we now explicitly state at line 55:
>
> > “Despite substantial progress, these methods rely on rendered images as intermediate carriers in the watermark embedding and extraction process, thereby protecting Gaussian primitives only indirectly; we refer to this protection paradigm as *indirect watermarking*.”
>
> In addition, at line 68 we now clarify:
>
> > "To avoid the risks associated with extracting watermarks from rendered images, we aim to develop a *native watermarking* framework that dispenses with rendered images as intermediaries and operates directly on Gaussian primitives."
>
> Compared with indirect watermarking, native watermarking removes the image as a relay medium, allowing a better balance between rendering quality and watermark accuracy. More importantly, native watermarking enables fine-grained local protection, which makes it better suited for defending against partial infringement. We hope that these revisions make the conceptual distinctions clearer and facilitate a more direct and fair comparison between NGS-Marker and existing methods.
>
> ---
>
> **W2. The definition of Partial Infringement also needs to be referred and compared with terms used in the literature.**
>
> **Response to W2:** In Section 3 (Problem Analysis), we formally introduce the notion of "Partial Infringement." To the best of our knowledge, there is currently no prior work in the 3DGS watermarking literature that investigates the same or a closely related problem, and therefore we did not cite directly relevant references in the original manuscript. In the revised version, we have updated the Related Work section by adding an analogy to watermarking methods for meshes [1] and point clouds [2] that are robust to large-scale cropping, in order to improve the overall coherence of the paper. Although these methods can, to some extent, extract a watermark from a local region of the protected asset, they are designed for scenarios where the entire 3D asset undergoes severe cropping, rather than the scenario we focus on, where a portion of a protected 3D asset is embedded into another asset.
>
> [1] Narendra M, Valarmathi M L, Anbarasi L J, et al. Levenberg–Marquardt deep neural watermarking for 3D mesh using nearest centroid salient point learning[J]. Scientific Reports, 2024, 14(1): 6942.
>
> [2] Zaman K A U, Alam M Z, Ali M N M, et al. Deep Neural Watermarking for Robust Copyright Protection in 3D Point Clouds[J]. arXiv preprint arXiv:2510.27533, 2025.
>
> ---
>
> **W4. Injecting image-based watermarks into 3D scenes using NGS-Marker is okay but not very personalized.**
>
> **Response to W4:** Thank you for reviewing our manuscript. Our intention in Sec. 4.4 and Sec. 5.5 is to demonstrate that NGS-Marker is not restricted to fixed binary IDs and can also embed multimodal messages, including user-chosen images. In practice, these images can be **identity-aware assets such as a creator’s logo, avatar, or stylized signature**, so each rights holder can register a distinct secret image while still benefiting from our native 3D robustness and local decodability. To describe this more accurately and avoid confusion, we have revised the manuscript by replacing expressions related to "Personalized Watermark" with "Image-based Watermarking", and we only retain at line 101 a brief explanation that our method can support image-based personalized watermarking.

---

> ### Author Response · Authors · 2025-11-26
> **Response to W3 (1/2)**
>
> **W3. The performance comparison between the proposed method with three baselines is not enough and the performance analysis in robustness is very limited.**
>
> **Response to W3:** We sincerely thank you for your thoughtful review. In the revised manuscript, we have added and analyzed the robustness of our method against adversarial attacks as well as 3D editing in App. L. In addition, we have included further experimental results on complex scenes and dynamic scenes in App. M.
>
> **(1) Regarding adversarial attacks:** Following Kerckhoffs’ principle and the C&W (Carlini & Wagner) attack [3], we assume that the attacker knows all details of the watermarking algorithm but does not know the hyperparameters used during watermark embedding (e.g., `bit_len` and $\delta$), and that the message extractor remains private. In the attack process, the attacker first trains a message extractor, then sets the target message to 0.5, and finally uses gradient descent to optimize the watermarked scene so that the decoded message approaches the target message:
>
> $\mathcal{L} _ {\text{attack}}=\mathrm{BCE}(\mathcal{M} _ \text{attack}, \mathcal{M} _ {\text{target}}) + \mathrm{MSE}(\mathcal{R}(\mathcal{G} ^ {w}), \mathcal{R}(\mathcal{G} ^ {s})),$
>
> where, $\mathcal{M} _ \text{attack}$ denotes the watermark message extracted from the attacked scene by the attacker’s extractor, and $\mathcal{M} _ \text{target}$ is a target message sequence whose entries are all 0.5. During the attack, for the HiDDeN extractor used in GaussianMarker and 3D-GSW, we train the extractor on 10,000 images sampled from COCO; for GuardSplat, we retrain the extractor using the algorithms in their public codebase. For fair comparison, we fix the number of attack iterations to 300 for all methods. We conduct attacks on the "*bear*" scene embedded with a 16-bit message ($\delta$ = 8192), and the experimental results of Bit-Acc are shown in the table below.
>
> | Attacker     | 8 bits $\delta$:8192 | 16 bits $\delta$:4096 | 16 bits $\delta$:8192 | 24 bits $\delta$:8192 | Avg.  |
> |----------------|:---------------:|:----------------:|:----------------:|:----------------:|:-------:|
> | 3D-GSW         | 57.29         | 58.55          | 58.55          | 67.19          | 60.40 |
> | GaussianMarker | 57.74         | 57.39          | 57.39          | 63.30          | 59.00 |
> | GuardSplat     | 62.03         | 65.42          | 65.42          | 72.51          | 66.35 |
> | NGS-Marker     | 93.50         | 79.22          | 65.83          | 92.16          | 82.68 |
>
> We observe that NGS-Marker and the baseline methods exhibit clearly different behaviors under adversarial attacks. When the attacker does not correctly guess the hyperparameters we use during watermark embedding, our method is able to effectively withstand the attack; only when the hyperparameters are exactly matched does the watermark accuracy of our method drop significantly. In contrast, for existing 3DGS watermarking methods that rely on rendered images for indirect protection, different hyperparameter choices already cause severe degradation of the embedded watermark. We have added this set of experiments to App. L.
>
> **(2) Regarding 3D editing:** We utilize GaussianEditor [4] to edit the watermarked scenes and then detect the watermark information in the edited scenes. Specifically, we select two scenes, "*bear*" and "*person*", and apply the editing prompts "*Turn the bear into a grizzly bear*" and "*Turn him into a clown,*" respectively. The Bit-Acc results are reported in the Table below. As can be seen, our method is affected the least even under 3D edits that significantly alter object appearance, whereas the accuracy of existing indirect watermarking approaches drops substantially.
>
> | Bit-Acc       | *"Turn the bear into a grizzly bear"* | *"Turn him into a clown"* |
> |---------------|:-----------------------------------:|:------------------------:|
> | 3D-GSW        | 72.40                             | 74.39                  |
> | GaussianMarker| 70.26                             | 69.42                  |
> | GuardSplat    | 78.55                             | 77.18                  |
> | NGS-Marker    | 91.06                             | 90.53                  |
>
>
> [3] Carlini N, Wagner D. Towards evaluating the robustness of neural networks[C]//2017 ieee symposium on security and privacy (sp). Ieee, 2017: 39-57.
>
> [4] Chen Y, Chen Z, Zhang C, et al. Gaussianeditor: Swift and controllable 3d editing with gaussian splatting[C]//Proceedings of the IEEE/CVF conference on computer vision and pattern recognition. 2024: 21476-21485

---

> ### Author Response · Authors · 2025-11-26
> **Response to W3 (2/2)**
>
> **(3) Regarding complex scenes:** We have further evaluated our method on four scenes in the T&T dataset ("*train*", "*truck*", "*drjohnson*", and "*playroom*"), and the results are shown in the table below. As can be seen, our approach is able to produce stable watermarking effects even in these complex scenarios.
>
> | Scene | 8 bits Bit-Acc | 8 bits PSNR | 16 bits Bit-Acc | 16 bits PSNR | 24 bits Bit-Acc | 24 bits PSNR |
> |-|:-:|:-:|:-:|:-:|:-:|:-:|
> | *Train* | 98.75 | 41.07       | 97.83 | 40.12        | 94.50           | 39.05        |
> | *Truck* | 99.03  | 41.00       | 97.84 | 39.61        | 94.73           | 39.20        |
> | *Drjohnson* | 98.06  | 39.48 | 96.46           | 39.13        | 93.22           | 38.43        |
> | *Playroom* | 98.32          | 40.19       | 97.59           | 39.38        | 93.65           | 38.71        |
>
> **(4) Regarding dynamic scenes:** To represent dynamic scenes, we adopt the classical 4DGS method [5]. In 4DGS, the primitives at any time step $t$ are not explicitly stored but are predicted by the `deform_network`. Therefore, during watermark embedding we also optimize the weights of the `deform_network`, with the objective that any local region of the decoded 3D scene at any time step should contain the complete watermark information:
>
> $\mathcal{L} _ {\text{dynamic}} = \mathrm{BCE}(\mathcal{M} ^ {\text{id}}, \mathcal{E}(\text{Sample}(\mathcal{D}(\mathcal{G} ^ w, t)))) + \mathrm{MSE}(\mathcal{R}(\mathcal{D}(\mathcal{G} ^ {w},t)), \mathcal{R}(\mathcal{D}(\mathcal{G} ^ {s},t))),$
>
> where, $\mathcal{E}$ denotes the message extractor, Sample denotes the local Gaussian primitive sampling operation, $\mathcal{D}$ denotes the `deform_network`, and $t$ is the time step in the dynamic scene. We conduct experiments on the D-NeRF dataset, and the results are shown in the table below (for ease of comparison, we also list in parentheses the corresponding results in static scenes). As can be seen, our method can be almost seamlessly extended to dynamic scenes.
>
> | Bits | Bit-Acc | PSNR↑  | SSIM↑  | LPIPS↓  |
> |-|:-:|:-:|:-:|:-:|
> | 8 bits | 98.71 (99.26) | 41.50 (41.77) | 0.996 (0.996) | 0.004 (0.004) |
> | 16 bits| 98.24 (98.35) | 40.29 (40.17) | 0.995 (0.995) | 0.008 (0.007) |
> | 24 bits| 95.22 (94.79) | 39.48 (39.61) | 0.993 (0.993) | 0.013 (0.013) |
>
> [5] Wu G, Yi T, Fang J, et al. 4d gaussian splatting for real-time dynamic scene rendering[C] // Proceedings of the IEEE/CVF conference on computer vision and pattern recognition. 2024: 20310-20320.

---

> ### Author Response · Authors · 2025-11-26
> **Response to Q1, Q2, and Q3**
>
> **Q1. What mostly make the proposed watermarking framework NGS-Maker different from the existing methods?**
>
> **Response to Q1:** In existing methods, both watermark embedding and extraction **require rendering images as an intermediate carrier to indirectly protect the Gaussian primitives**. This causes their accuracy to drop sharply when the appearance of the protected asset shifts, and using rendered images as the carrier inevitably degrades the visual quality. In our work, **the proposed native watermarking scheme avoids any reliance on rendered images during both embedding and extraction, and instead operates directly on the Gaussian primitives**. As a result, watermark detection is no longer affected by the distribution of rendered images, and watermark embedding can be achieved while better preserving rendering quality. More importantly, the native watermarking design allows us to protect only local subsets of Gaussian primitives, thereby providing effective resistance against partial infringement.
>
> ---
>
> **Q2. What are the concrete concepts/definitions of “indirect watermarking” and “native watermarking”?**
>
> **Response to Q2:** We would like to thank you again for your review. For a detailed response to this question, please refer to **Response to W1**.
>
> ---
>
> **Q3. What are the methods in the literature most similar to the proposed one?**
>
> **Response to Q3:**  The methods most closely related to our work are the three baselines compared in our experiments (3D-GSW [6], GaussianMarker [7], and GuardSplat [8]). **Our native watermarking approach is mainly designed to address a common limitation shared by these methods: their reliance on rendered-image-based, indirect watermarking strategies, which are inherently less robust and prone to introducing undesired artifacts into the rendered results.**
>
> We sincerely hope that our responses and the corresponding revisions to the manuscript have addressed your concerns. If you still have any suggestions or questions, we would be very grateful for the opportunity to continue the discussion and further improve our work based on your valuable feedback.
>
> [6] Jang Y, Park H, Yang F, et al. 3d-gsw: 3d gaussian splatting for robust watermarking[C]//Proceedings of the Computer Vision and Pattern Recognition Conference. 2025: 5938-5948.
>
> [7] Huang X, Li R, Cheung Y, et al. Gaussianmarker: Uncertainty-aware copyright protection of 3d gaussian splatting[J]. Advances in Neural Information Processing Systems, 2024, 37: 33037-33060.
>
> [8] Chen Z, Wang G, Zhu J, et al. GuardSplat: Efficient and Robust Watermarking for 3D Gaussian Splatting[C]//Proceedings of the Computer Vision and Pattern Recognition Conference. 2025: 16325-16335.

---

### Official Review · Reviewer_FtXd · 2025-10-31

**Soundness:** 3
**Presentation:** 3
**Contribution:** 3
**Rating:** 6
**Confidence:** 3

**Summary:**

NGS-Marker proposes a native 3DGS watermark that is embedded and decoded directly from local Gaussian neighborhoods via a progressive optimization pass. The method targets partial infringement (copying only subsets of Gaussians) and reports strong per-Gaussian detection and rendering fidelity, plus optional hybrid protection with indirect, image-domain marks and a proof-of-concept for personalized watermarks.

**Strengths:**

1. Composable design. The proposed hybrid objective indicates native marks don’t conflict with rendering-domain watermarks, and the image-message experiment shows the architecture can accommodate non-binary payloads with small changes to encoders/decoders. This extends the method’s use beyond simple IDs.

2. Minimal visual impact with qualitative and quantitative evidence. Difference maps show small, spatially diffuse changes, and PSNR/SSIM/LPIPS metrics remain strong after embedding; this is the right mix of perceptual and pixel-wise indicators for vision papers. The visual comparisons make the progressive optimization design choice more convincing.

3. Clear pipeline and rationale. The progression from local injection to progressive, scene-level embedding is well-motivated: naive block-wise injection yields boundary artifacts, whereas repeated random injection accumulates distortion; the final procedure addresses both issues. This kind of explicit failure-mode discussion is useful for re-implementation.

**Weaknesses:**

1. White-box resilience not evaluated. The attack suite focuses on stochastic distortions rather than adversarial removal with knowledge of the extractor. With the absence of such results, the robustness to an adaptive attacker remains an open question.

2. Scaling cost and production fit. Embedding time rises from 4 minutes (42k primitives) to 35 minutes (589k), which is acceptable for small scenes but may be significant for game/film assets with millions of Gaussians; the paper notes scalability opportunities but does not provide an algorithmic path or empirical stress test beyond the four test scenes.

**Questions:**

1. Verification cost: What is end-to-end verification latency (per million Gaussians), including neighborhood sampling and similarity computation? Any heuristics to prune the search while maintaining recall?

2. Breadth of validation: Could you add evaluations on larger, diverse benchmarks (e.g., Tanks & Temples) and any dynamic/4DGS content to assess generalization?

---

> ### Author Response · Authors · 2025-11-26
> **Response to W1 and W2**
>
> **W1. White-box resilience not evaluated. The attack suite focuses on stochastic distortions rather than adversarial removal with knowledge of the extractor.**
>
> **Response to W1:** Thank you very much for pointing out this important metric. Since existing 3DGS watermarking works do not consider adversarial attacks, we designed a dedicated adversarial attack pipeline tailored to 3DGS watermarking methods. Following Kerckhoffs’ principle and the C&W (Carlini & Wagner) attack [1], we assume that the attacker knows all details of the watermarking algorithm but does not know the hyperparameters used during watermark embedding (e.g., `bit_len` and $\delta$), and that the message extractor remains private. In the attack process, the attacker first trains a message extractor, then sets the target message to 0.5, and finally uses gradient descent to optimize the watermarked scene so that the decoded message approaches the target message:
>
> $\mathcal{L} _ {\text{attack}}=\mathrm{BCE}(\mathcal{M} _ \text{attack}, \mathcal{M} _ {\text{target}}) + \mathrm{MSE}(\mathcal{R}(\mathcal{G} ^ {w}), \mathcal{R}(\mathcal{G} ^ {s})),$
>
> where, $\mathcal{M} _ \text{attack}$ denotes the watermark message extracted from the attacked scene by the attacker’s extractor, and $\mathcal{M} _ \text{target}$ is a target message sequence whose entries are all 0.5. During the attack, for the HiDDeN extractor used in GaussianMarker and 3D-GSW, we train the extractor on 10,000 images sampled from COCO; for GuardSplat, we retrain the extractor using the algorithms in their public codebase. For fair comparison, we fix the number of attack iterations to 300 for all methods. We conduct attacks on the "*bear*" scene embedded with a 16-bit message ($\delta$ = 8192), and the experimental results of Bit-Acc are shown in the table below.
>
> | Attacker     | 8 bits $\delta$:8192 | 16 bits $\delta$:4096 | 16 bits $\delta$:8192 | 24 bits $\delta$:8192 | Avg.  |
> |----------------|:---------------:|:----------------:|:----------------:|:----------------:|:-------:|
> | 3D-GSW         | 57.29         | 58.55          | 58.55          | 67.19          | 60.40 |
> | GaussianMarker | 57.74         | 57.39          | 57.39          | 63.30          | 59.00 |
> | GuardSplat     | 62.03         | 65.42          | 65.42          | 72.51          | 66.35 |
> | NGS-Marker     | 93.50         | 79.22          | 65.83          | 92.16          | 82.68 |
>
>
> We observe that NGS-Marker and the baseline methods exhibit clearly different behaviors under adversarial attacks. When the attacker does not correctly guess the hyperparameters we use during watermark embedding, our method is able to effectively withstand the attack; only when the hyperparameters are exactly matched does the watermark accuracy of our method drop significantly. In contrast, for existing 3DGS watermarking methods that rely on rendered images for indirect protection, different hyperparameter choices already cause severe degradation of the embedded watermark. We have added this set of experiments to App. L.
>
> [1] Carlini N, Wagner D. Towards evaluating the robustness of neural networks[C]//2017 ieee symposium on security and privacy (sp). Ieee, 2017: 39-57.
>
> ---
>
> **W2. Scaling cost and production fit.**
>
> **Response to W2:** In App. J of the revised manuscript, we provide a concrete and simple parallelization strategy to accelerate watermark embedding, together with corresponding experiments. Specifically, we uniformly partition a 3D scene into four regions along the x–y plane, use four GPUs to embed watermarks into these regions in parallel, and then recombine the four parts followed by a second optimization on the merged scene. As shown in the table below, the parallel implementation substantially reduces the time required for watermark embedding. It is reasonable to expect that, for very large scenes, as long as a sufficient number of GPUs are available to support block-wise watermarking, the embedding time can be maintained at a relatively low level.
>
> Time (min) | Person | Chair | Bear | Garden | Avg.
> --- | :---: | :---: | :---: | :---: | :---:
> NGS-Marker | 4.0 | 9.3 | 28.7 | 35.2 | 19.3
> NGS-Marker-Parallel | 2.7 | 4.6 | 12.5 | 13.9 | 8.4
>
> The feasibility of such parallel acceleration essentially stems from our proposed “**native watermarking**” idea. Existing 3DGS-based watermarking methods rely on rendered images as an intermediate both during watermark embedding and detection; however, different rendered images may still be associated with the same Gaussian primitive, which makes it difficult for these methods to be efficiently parallelized. In contrast, since our method operates directly on the Gaussian primitives, it is naturally compatible with parallel optimization strategies.

---

> ### Author Response · Authors · 2025-11-26
> **Response to Q1 and Q2**
>
> **Q1. Verification cost: What is end-to-end verification latency (per million Gaussians), including neighborhood sampling and similarity computation? Any heuristics to prune the search while maintaining recall?**
>
> **Response to Q1:** On a single NVIDIA A100 GPU, extracting 8,192 neighboring Gaussian primitives from the candidate scene and performing watermark detection and comparison takes approximately 21ms in our current implementation. For a scene containing one million Gaussian primitives, a theoretical non-overlapping sampling that covers all primitives would therefore require roughly 3 seconds to complete a full-coverage detection. However, to achieve the precise localization results shown in Figure 4 of our paper, we perform multiple overlapping samplings for each primitive and then compute a weighted average of the detection scores. In our experiments, we observed that repeating the coverage sampling 60–80 times typically yields satisfactory visual results. This implies that, on a single A100 GPU, it takes about 3–4 minutes to accurately localize the plagiarized regions in a scene with one million primitives. Since each detection is independent, parallel watermark detection is in principle feasible, and the overall detection time can be further reduced accordingly.
>
> As heuristic strategies, we suggest: (1) first uniformly sampling several sets of Gaussian primitives over the whole scene, then examining whether high bit-acc (bit accuracy) values tend to concentrate in certain regions, and finally focusing detection on these high–bit-acc areas; (2) if the protected data are relatively familiar to the user, one may first visually inspect the entire scene to identify suspected plagiarized regions, and then prioritize detection on these regions while reducing the number of computations in other areas.
>
> ---
>
> **Q2. Breadth of validation: Could you add evaluations on larger, diverse benchmarks (e.g., Tanks & Temples) and any dynamic/4DGS content to assess generalization?**
>
> **Response to Q2:** Thank you very much for your helpful suggestion. In the revised manuscript, we have updated App. M to include additional experimental results on four scenes from the T&T dataset ("*train*", "*truck*", "*drjohnson*", "*playroom*"), as well as results on dynamic scenes.
>
> **(1) Regarding the T&T dataset:** The results on these four scenes are reported in the table below. We observe that the accuracy of our method on these scenes is close to the original evaluation results, indicating that our approach maintains consistent performance under these additional settings.
>
> | Scene | 8 bits Bit-Acc | 8 bits PSNR | 16 bits Bit-Acc | 16 bits PSNR | 24 bits Bit-Acc | 24 bits PSNR |
> |-|:-:|:-:|:-:|:-:|:-:|:-:|
> | *Train* | 98.75 | 41.07       | 97.83 | 40.12        | 94.50           | 39.05        |
> | *Truck* | 99.03  | 41.00       | 97.84 | 39.61        | 94.73           | 39.20        |
> | *Drjohnson* | 98.06  | 39.48 | 96.46           | 39.13        | 93.22           | 38.43        |
> | *Playroom* | 98.32          | 40.19       | 97.59           | 39.38        | 93.65           | 38.71        |
> | Original Test Results | 99.26          | 41.77       | 98.35           | 40.17        | 94.79           | 39.61        |
>
> **(2) Regarding dynamic scenes:** To represent dynamic scenes, we adopt the classical 4DGS method [2]. In 4DGS, the primitives at any time step $t$ are not explicitly stored but are predicted by the `deform_network`. Therefore, during watermark embedding we also optimize the weights of the `deform_network`, with the objective that any local region of the decoded 3D scene at any time step should contain the complete watermark information:
>
> $\mathcal{L} _ {\text{dynamic}} = \mathrm{BCE}(\mathcal{M} ^ {\text{id}}, \mathcal{E}(\text{Sample}(\mathcal{D}(\mathcal{G} ^ w, t)))) + \mathrm{MSE}(\mathcal{R}(\mathcal{D}(\mathcal{G} ^ {w},t)), \mathcal{R}(\mathcal{D}(\mathcal{G} ^ {s},t))),$
>
> where, $\mathcal{E}$ denotes the message extractor, Sample denotes the local Gaussian primitive sampling operation, $\mathcal{D}$ denotes the `deform_network`, and $t$ is the time step in the dynamic scene. We conduct experiments on the D-NeRF dataset, and the results are shown in the table below (for ease of comparison, we also list in parentheses the corresponding results in static scenes). As can be seen, our method can be almost seamlessly extended to dynamic scenes.
>
> We sincerely hope these additional experiments adequately address your concerns.
>
> | Bits | Bit-Acc | PSNR↑  | SSIM↑  | LPIPS↓  |
> |-|:-:|:-:|:-:|:-:|
> | 8 bits | 98.71 (99.26) | 41.50 (41.77) | 0.996 (0.996) | 0.004 (0.004) |
> | 16 bits| 98.24 (98.35) | 40.29 (40.17) | 0.995 (0.995) | 0.008 (0.007) |
> | 24 bits| 95.22 (94.79) | 39.48 (39.61) | 0.993 (0.993) | 0.013 (0.013) |
>
>
> [2] Wu G, Yi T, Fang J, et al. 4d gaussian splatting for real-time dynamic scene rendering[C] // Proceedings of the IEEE/CVF conference on computer vision and pattern recognition. 2024: 20310-20320.

---

### Official Review · Reviewer_ERh9 · 2025-11-01

**Soundness:** 3
**Presentation:** 3
**Contribution:** 3
**Rating:** 6
**Confidence:** 3

**Summary:**

This paper introduces NGS-Marker, a novel native watermarking framework designed to protect 3D Gaussian Splatting assets against partial infringement. The authors identify a critical limitation in existing indirect watermarking methods, which fail when the rendering distribution changes significantly. NGS-Marker addresses this by embedding watermarks directly into the Gaussian primitives using a jointly trained injector-extractor pair and a progressive gradient-based optimization strategy. The method supports hybrid protection and multimodal watermarking, and demonstrates robustness to common distortions.

**Strengths:**

1. The paper clearly identifies and formalizes the partial infringement problem in 3DGS, which is both practical and underexplored. This is a meaningful contribution to the field of 3D asset protection.

2. The proposed framework enable native local watermarking for 3DGS. The use of a perturbation-based injector and extractor, combined with progressive optimization, is well-motivated and technically sound.

**Weaknesses:**

1. Although the method supports large scenes, the embedding time grows with the number of primitives. Is it possible to add a comparison of watermark embedding time and extracting time with previous methods in the experiment?


2. Lack of discussion and comparison on computational complexity. Since the proposed method utilizes Point Transformer as the injector, adding experimental results on this aspect would better reflect the model's characteristics.

**Questions:**

1. How does this method perform in areas with significant overlap, such as when the embedded Gaussians and the watermarked Gaussians interweave with each other?

2. Could the choice of $\delta$ influence the noise robustness of the proposed method? Especially for densification and dropout.

---

> ### Author Response · Authors · 2025-11-26
> **Response to W1 and W2**
>
> **W1. Is it possible to add a comparison of watermark embedding time and extracting time with previous methods in the experiment?**
>
> **Response to W1:** We sincerely appreciate your constructive comments. In the revised manuscript, we have added a comparison of the watermarking time of our method and the baselines across different scenes in App. I, as summarized in the table below. Thanks to our native watermarking scheme that directly operates on Gaussian primitives, our method can parallelize the watermark embedding process by partitioning large scenes into smaller tiles. In contrast, existing 3DGS watermarking methods have to decode the watermark from rendered images, where the Gaussian primitives affected by different rendered views may overlap, making efficient parallel computation difficult. In the table, NGS-Marker-Parallel denotes the time cost of our method when the entire scene is evenly divided into four tiles along the xy-plane and watermarks are embedded in parallel. As can be seen, the watermark embedding time of all methods generally increases with scene complexity, while our method achieves the shortest average embedding time when the parallelization strategy is used.
>
> Time (min) | Person | Chair | Bear | Garden | Avg.
> --- | :---: | :---: | :---: | :---: | :---:
> 3D-GSW | 12.8 | 9.5 | 21.4 | 27.7 | 17.9
> GaussianMarker | 4.2 | 4.3 | 12.7 | 16.5 | 9.4
> GuardSplat | 5.5 | 4.9 | 15.2 | 19.5 | 11.3
> NGS-Marker | 4.0 | 9.3 | 28.7 | 35.2 | 19.3
> NGS-Marker-Parallel | 2.7 | 4.6 | 12.5 | 13.9 | 8.4
>
> In addition, we have evaluated the time required for a single watermark detection. For existing 3DGS watermarking strategies, watermark detection involves both image rendering and message extraction, whereas for our method, detection involves extracting a subset of Gaussian primitives followed by message extraction. We report in the table the time required by each method to perform one detection on a single A100 GPU. The results show that the watermark extraction time of our method is comparable to that of existing methods, while the additional time overhead incurred by our method grows more slowly as the scene becomes more complex. This confirms that NGS-Marker remains efficient and scalable for large-scale 3DGS assets, satisfying practical deployment requirements.
>
> Time (ms) | Person | Chair | Bear | Garden | Avg.
> --- | :---: | :---: | :---: | :---: | :---:
> 3D-GSW | 16.2 |17.2 |18.7 |19.1 |17.8
> GaussianMarker | 15.0 |16.1 |17.5 |18.0 |16.7
> GuardSplat | 17.2 |18.3 |19.7 |20.2 |18.9
> NGS-Marker | 20.7 |20.7 |20.9 |21.0 |20.8
>
> ---
>
> **W2. Lack of discussion and comparison on computational complexity.**
>
> **Response to W2:** Thank you very much for pointing this out. We agree that a more explicit analysis of the computational complexity will help clarify the characteristics of our method. Although we adopt a Point Transformer, the computational cost of watermark embedding and extraction does not increase sharply, mainly for two reasons: (1) our method does not require feeding the entire scene into the Point Transformer; instead, we only process a fixed number of primitives at a time; (2) before feeding the Gaussian primitives into the Point Transformer, we use FPS and KNN to convert them into a small number of tokens, which significantly reduces the attention cost inside the transformer.
>
> To more intuitively compare the computational complexity of our method with existing approaches, we measured both the number of parameters and the computational cost for a single forward pass. For other methods that extract watermarks from images, we fixed the input image size to 224×224. For our method, we fixed the number of Gaussian primitives to 8192. We used the open-source tools *thop* and *fvcore* to measure the parameter count and computation, respectively, and the results are summarized in the table below. As can be seen, CNN-based methods (3D-GSW, GaussianMarker) have relatively few parameters but a higher computational cost, whereas transformer-based methods (GuardSplat, NGS-Marker) have more parameters but a lower computational cost. Compared with these existing methods, the computational complexity of our approach does not increase significantly.
>
> Thank you again for this helpful suggestion, we have added the corresponding discussion to App. J. We hope this clarification and the additional empirical results adequately address your concern about computational complexity.
>
> Method | Params (M) | FLOPs (G)
> --- | :---: | :---:
> 3D-GSW | 0.29 | 14.48
> GaussianMarker | 0.29 | 14.48
> GuardSplat | 151.48 | 4.37
> NGS-Marker | 51.28 | 6.61

---

> ### Author Response · Authors · 2025-11-26
> **Response to Q1 and Q2**
>
> **Q1. How does this method perform in areas with significant overlap, such as when the embedded Gaussians and the watermarked Gaussians interweave with each other?**
>
> **Response to Q1:** Thank you for raising this very interesting question. We have added the corresponding analysis in App. G of the revised manuscript. Specifically, we gradually merge “*bear*” with embedded Gaussians to construct overlapping cases. Through visual analysis, we observe that **our method can consistently and accurately capture visually discernible infringing regions.** For regions that are not visible in the rendered image, infringement becomes meaningless, and we therefore do not pursue further analysis.
>
> **Q2. Could the choice of $\delta$ influence the noise robustness of the proposed method? Especially for densification and dropout.**
>
> **Response to Q2:** In App. K of the revised manuscript, we conduct an ablation study on the influence of $\delta$ on robustness. We vary the value of $\delta$ and then measure the Bit-Acc of watermarked scenes under different types of perturbations; the results are summarized in the table below (we use subscripts to denote the decrease in accuracy under densification and dropout). We observe that, as $\delta$ decreases, the overall robustness of the watermark is slightly reduced; however, within a reasonable range, changing $\delta$ has only a modest impact on robustness. After a scene is perturbed, the choice of $\delta$ also affects the number of perturbed primitives that are fed into the message extractor. When $\delta$ is smaller, although the detection system appears more fragile, each detection is exposed to fewer perturbations. Consequently, for scenes subjected to the same level of perturbation, the robustness of the watermark corresponding to different values of $\delta$ does not differ significantly.
>
> | $\delta$     | None | Noise ($\sigma=0.015$) | Rotation ($\pm\pi$) | Scaling ($\pm\infty$) | Densification (0%-50%) | Dropout (0%-50%) | Translation ($\pm\infty$) |
> |-------|:------:|:--------------------------:|:---------------:|:--------------:|:------------------------:|:------------------:|:------------------:|
> | 2048  | 95.62 | 93.89 | 95.62 | 95.62 | 94.14$_{-1.48}$ | 93.78$_{-1.84}$ | 95.62 |
> | 4096  | 96.77 | 95.19 | 96.77 | 96.77 | 95.83$_{-0.94}$ | 94.94$_{-1.83}$ | 96.77 |
> | 6144  | 96.83 | 95.30 | 96.83 | 96.83 | 96.22$_{-0.61}$ | 95.36$_{-1.47}$ | 96.83 |
> | 8192  | 98.35 | 97.06 | 98.35 | 98.35 | 97.93$_{-0.42}$| 97.41$_{-0.94}$ | 98.35 |
> | 12288 | 98.17 | 97.02 | 98.17 | 98.17 | 97.82$_{-0.35}$ | 97.26$_{-0.91}$ | 98.17 |

---

### Comment · Area_Chair_D1cQ · 2025-11-25

Dear Reviewers,

Thank you for your time and effort in reviewing submissions for ICLR 2026. As we begin the author-reviewer discussion process, we kindly remind you to submit your responses to the author rebuttals by **December 2**.

Your engagement in this discussion phase is crucial to ensuring a fair and thorough evaluation of each submission.

### **Action Required**
- Carefully consider the authors’ rebuttal and any additional evidence they provide.
- Update your review (if applicable) to reflect your revised perspective.
- Discuss with the authors if further details are required

Your AC

---

### Author Response · Authors · 2025-11-26

Dear Reviewers,

We sincerely thank you for your valuable feedback and constructive suggestions. Based on your comments, we have enhanced the paper as follows:

**Experiments & Analysis**
- [Reviewer ERh9, FtXd, 5NBU] We have reported a comparison of the time required for watermark embedding and extraction between our method and prior approaches, and we have described a concrete parallelization strategy to accelerate watermark embedding.
- [Reviewer ERh9] We have provided a comparison and analysis of the computational complexity.
- [Reviewer ERh9] We have investigated watermark detection when watermarked and non-watermarked Gaussian primitives overlap in space.
- [Reviewer ERh9] We have conducted ablation experiments to study the impact of $\delta$ on robustness.
- [Reviewer FtXd, NYv5, 5NBU] We have performed experiments to evaluate the robustness of NGS-Marker against adversarial attacks and 3D editing operations.
- [Reviewer FtXd, NYv5, 5NBU] We have further evaluated our method on complex scenes and dynamic scenes.
- [Reviewer 5NBU] We have added ablation experiments to investigate how using different Gaussian attributes affects the watermarking performance.

**Clarifications & Modifications**
- [Reviewer NYv5] We have provided more precise definitions of "**indirect watermarking**" and "**native watermarking**", and added relevant citations to improve the clarity and flow of the exposition.
- [Reviewer NYv5] We have revised the description related to “Personalized Watermark” to improve its accuracy.
- [Reviewer 5NBU] We have added a more theoretical analysis explaining why Gaussian primitives are suitable for watermark embedding.
- [Reviewer 5NBU] We have included additional details about the hybrid (cooperative) protection procedure.

We have uploaded the revised manuscript, in which all changes are highlighted in blue. We believe these revisions substantially strengthen the paper and hope they adequately address your concerns. Thank you again for your thoughtful and constructive reviews.

Sincerely,

The Authors

---

### Author Response · Authors · 2025-12-03
**Summary of Rebuttal and Revisions for Paper 2991**

Dear AC,

We sincerely appreciate your time and effort in managing this submission. We have uploaded the revised manuscript (changes marked in blue) and provided detailed responses to each reviewer’s comments. To facilitate your review, we briefly summarize the reviewers’ main points and our corresponding responses below.

**1. The reviewers acknowledged the merits of our work, including but not limited to:**
* **Clear definition of a critical vulnerability:**
    * Reviewer`ERh9`: *"The paper clearly identifies ... This is a meaningful contribution ..."*
    * Reviewer`5NBU`: *"... identifies a critical and underexplored vulnerability ... The motivation is clearly articulated."*
* **Innovative and sound architectural design:**
    * Reviewer`ERh9`: *"The proposed framework ... is well-motivated and technically sound."*
    * Reviewer`FtXd`: *"The progression ... is well-motivated."*
* **Flexibility and functional richness:**
    * Reviewer`FtXd`: *"This extends the method’s use beyond simple IDs."*
    * Reviewer`NYv5`: *"Hybrid protection and personalized watermarking are both supported."*
    * Reviewer`5NBU`: *"... demonstrate practical flexibility."*

**2. Summary of Responses and Revisions**

**(1) Reviewer`ERh9`(Score: 6)**
* **Time and Computational Complexity:** Relevant comparisons and discussions have been added in App. I and J. Results indicate that our method is on par with the baseline in runtime and computational complexity.
* **Interweaving Behavior:** In App. G, we provided visualization experiments demonstrating that our method can consistently and accurately capture visually discernible infringing regions.
* **Impact of $\delta$ on Robustness:** We conducted ablation studies in App. K, which show that changing $\delta$ within a reasonable range has only a modest impact on robustness.

**(2) Reviewer`FtXd`(Score: 6)**
* **Adversarial Attack:** We added adversarial attack experiments in App. L, showing that our method defends against attacks more effectively than baselines.
* **Scalability Algorithms:** In App. I, we proposed a practical parallel watermarking strategy and reported the time required, demonstrating that parallel computing significantly accelerates watermark injection.
* **Some Questions** (Verification Cost and More Validation): We have provided detailed answers in "Response to Q1 and Q2" and added relevant experiments to App. M.

**(3) Reviewer`NYv5`(Score: 2)**

Regarding reviewer`NYv5`'s concerns, we recognize that they mainly stem from insufficient clarity in our problem definition and motivation. This was likely due to the overly concise Introduction, which made our intent less accessible for some readers (reviewers`ERh9`and`5NBU`pointed out that the problem definition and motivation are clear). We thank reviewer`NYv5`for the valuable feedback and have revised the manuscript.

* **Definition of Indirect/Native Watermarking:** We have explicitly defined indirect and native watermarking in Lines 054 and 068, respectively. *The key distinction is whether the watermarking process needs rendered images as the intermediate medium.*
* **References for Partial Infringement:** We have added references in Related Work regarding point cloud and mesh watermarking that resist large-scale cropping (similarly emphasizing "partial" protection), along with comparisons.
* **More Experiments:** In the revised manuscript, we added results for complex and dynamic scenes (App. M), tested robustness against adversarial attacks and 3D editing (App. L), and provided further ablation studies (App. K).
* **Personalization Terminology:** We revised the manuscript by replacing expressions related to "Personalized Watermark" with "Image-based Watermarking."
* **Some Questions** (The biggest difference from existing methods? The concrete definitions of indirect/native watermarking? The most relevant work?): We have provided detailed responses to these questions one by one in "Response to Q1, Q2, and Q3".

**(4) Reviewer`5NBU`(Score: 6)**
* **Analysis of Gaussian Primitives:** We added a detailed analysis in App. N regarding which properties make Gaussian primitives suitable for watermark embedding.
* **Partition and Traversal Strategy:** In App. I, we presented a practical parallel acceleration strategy.
* **Robustness (Adversarial/Editing) and Impact of Different Attributes:** We added relevant experiments in App. L  and K.
* **Some Questions (FPR, More Details, Dynamic Scenes, etc.):** We responded to these questions point-by-point in "Response to Q1, Q2, Q3, and Q4".

**Final Remark**

Overall, our work received positive evaluations from most reviewers (3/4). We have systematically revised the paper to clarify the problem definition, strengthen the connection to existing literature, and add more experiments. We believe these revisions sufficiently address the concerns of reviewer `NYv5` and the other reviewers.

Once again, thank you for your time and consideration.

Sincerely,

The Authors

---

### Meta-Review · Area_Chair_QKSL · 2026-01-04

**Summary:**

This paper initially received three positive reviews and one negative review. The negative review primarily raised concerns about insufficient clarification of specific concepts and terminology, such as the distinction between native and indirect watermarking, as well as a lack of detailed explanations in parts of the manuscript. I consider these issues to be primarily related to presentation and exposition rather than fundamental technical flaws, and they can be addressed with relatively minor revisions. The authors addressed these concerns in the rebuttal and provided clarifications that I find largely adequate.

Overall, the authors have addressed most of the reviewers' concerns raised during the rebuttal. The remaining issues from the negative review were primarily related to clarity and presentation, and these have been adequately resolved. The paper demonstrates that watermarking can be embedded directly into 3D primitives and provides sufficient experimental evidence. Based on the tone of the initial review, it is unlikely that NYv5 would revise its assessment to a positive score. However, I do not consider the identified weaknesses critical, and overall, I am leaning toward recommending acceptance.

Minor comments from AC (to be addressed in the final version):

1. Figure 2: The feasibility analysis appears only loosely connected to the central theme of the paper, and its relevance is not entirely clear.

2. Eq. (1): The final term P_g(tokenizer, f_m) does not seem necessary.

3. L147: _i^N -> _{i=1}^N

4. L148: s_i is not defined as a matrix.

**Reviewer Concerns:**

ERh9: The primary concerns relate to computational complexity. These issues were addressed during the rebuttal, and the responses appear largely satisfactory.

FtXd: This review raised concerns regarding white-box attack robustness, scalability, and computational complexity. The rebuttal partially addresses these points, particularly regarding scalability and efficiency. Robustness under strong white-box attacks remains limited. However, this may be considered beyond the scope of the current paper.

NYv5: The main criticisms focus on clarity, terminology, and conceptual distinctions (e.g., native vs. indirect watermarking). These are largely presentation-related issues that can be resolved with careful revision. The authors provided appropriate clarifications during the rebuttal.

5NBU: This reviewer raised concerns overlapping with other reviews, including white-box attack robustness and computational complexity, as well as the lack of theoretical analysis supporting Figure 1. While the rebuttal does not fully resolve the request for theoretical justification, I do not view this limitation as critical to the overall contribution.

**Reviewer Scores:**

See the above sections.

---

### Decision · Program_Chairs · 2026-01-26

Accept (Poster)